# Study on molecular orientation and stratification in RNA-lipid nanoparticles by cryogenic orbitrap secondary ion mass spectrometry

Anna M. Kotowska [1], Michael Fay[2], Julie A. Watts[2], Ian S. Gilmore[3], David J. Scurr [1], Alaina Howe[4], Vladimir Capka[4], Corey E. Perez [4], Devin Doud[4], Siddharth Patel [4], Mark Umbarger[4], Robert Langer [5,6] & Morgan R. Alexander [1] ✉

Lipid nanoparticle RNA (LNP-RNA) formulations are used for the delivery of vaccines and other therapies. RNA molecules are encapsulated within their interior through electrostatic interactions with positively charged lipids. The identity of the lipids that present at their surface play a role in how they interact with and are perceived by the body and their resultant potency. Here, we use a model formulation to develop cryogenic sample preparation for molecular depth profiling Orbitrap secondary ion mass spectrometry (Cryo-OrbiSIMS) preceded by morphological characterisation using cryogenic transmission electron microscopy (Cryo-TEM). It is found that the depth distribution of individual lipid components is revealed relative to the surface and the RNA cargo defining the core. A preferential lipid orientation can be determined for the 1,2-Dimyristoyl-glycero-3-methox-polyethylene glycol 2000 (DMG-PEG2k) molecule, by comparing the profiles of PEG to DMG fragments. PEG fragments are found immediately during analysis of the LNP surface, while the DMG fragments are deeper, coincident with RNA ions located in the core, in agreement with established models of LNPs. This laboratory-based de novo analysis technique requires no labelling, providing advantages over large facility neutron scattering characterisation.

Billions of doses of mRNA-loaded lipid nanoparticle (LNP) vaccines represent a spectacular recent example of the utility of therapeutic nanomaterials[1]. The molecules presented on the surface of such LNPs influence how the material interacts with the environment inside the body[2], including which biomolecules it adsorbs to form the biomolecular corona[3]. This *biointerface* is critical in determining nanoparticle trafficking, potency and immunological reaction, yet uncertainties surrounding LNP structural organisation, such as the location of ionisable lipids either in the core[4,5] or in the exterior shell, remain[6–9]. Such structural details likely have implications on nanoparticle behaviour, including the critical stage of endosomal escape in transfection, for which efficiencies are estimated to be as low as 2%[10,11].

Synthetic LNPs are commonly composed of four components; an ionisable lipid, a helper lipid, cholesterol and a poly(ethylene glycol)ylated

(PEG) lipid. Ionisable lipids interact electrostatically with the negatively charged mRNA facilitating its encapsulation within the LNP. These also enable endosomal escape by appropriate pKa choice. Helper lipids, such as the phospholipids distearoylphosphatidylcholine (DSPC) and dioleoyl-phosphatidylethanolamine (DOPE), contribute to the structural integrity and stability of LNPs. Cholesterol is incorporated into LNPs to modulate their fluidity and rigidity, enhancing particle stability and membrane fusion capabilities and also helps maintain the structural integrity of LNPs, allowing for delivery and release of mRNA into target cells. The inclusion of PEG-lipids on the LNP surface contributes to both formulation stability[11],and the 'stealth' properties of the nanoparticles[5,8], with longer fatty acids allowing them to circulate in the bloodstream longer and increasing the chances of reaching their target cells whilst shorter fatty acids are shed

[1]School of Pharmacy, University of Nottingham, Nottingham, UK. [2]Nanoscale and Microscale Research Centre, University of Nottingham, Nottingham, UK. [3]National Physical Laboratory, Hampton Rd, Teddington, UK. [4]Sail Biomedicines, Cambridge, MA, USA. [5]Department of Chemical Engineering, Massachusetts Institute of Technology, Cambridge, MA, USA. [6]Koch Institute for Integrative Cancer Research, Massachusetts Institute of Technology, Cambridge, MA, USA. ✉e-mail: morgan.alexander@nottingham.ac.uk

faster enabling the lipid surface to be exposed and absorb the serum proteins[12].

The study of the structural, compositional, and functional aspects of LNPs for drug delivery has employed various analytical and imaging techniques. Cryo-EM allows for the observation of the nanoparticle morphology and internal structure, providing insights into the lipid organisation, but without information on molecular identity[13,14]. Small-angle neutron scattering (SANS) combined with deuteration of components at large scale facilities has been used to quantify the lipid distribution at the surface of LNPs using isotopic contrast variation[5]. Small-angle X-ray scattering (SAXS) using synchrotron sources, has been used to probe bulk phase structures[15]. Dynamic light scattering (DLS) is used to rapidly measure the size distribution and polydispersity index (PDI) of LNPs, offering insights into their homogeneity and stability. NMR spectroscopy provides information on the molecular structure, dynamics, and interactions within LNPs[10]. It can be used to study the composition of the lipid core, the conformation of PEG chains, and the interaction between PEG and other components of LNPs, however the structural analysis of LNPs is limited by NMR sensitivity[16].

To provide a method that can address these uncertainties, here we show that molecular depth profiling of LNPs is possible using cryogenic OrbiTrap secondary ion mass spectrometry (Cryo-OrbiSIMS). We find this laboratory-based direct analysis method can identify the molecular structure of the LNP surface, revealing the relative depths of lipid and RNA components. Furthermore, the orientation of certain molecules can even be discerned where fragments resulting from different portions of the molecule are detected. This information relies on retention of the native hydrated structure within the vacuum of the Cryo-OrbiSIMS instrument, verified by analysis using cryo-transmission electron microscopy (Cryo-TEM) on the same sample.

Vacuum-based surface chemical analysis techniques such as X-ray photoelectron spectroscopy (XPS) or time of flight secondary ion mass spectrometry (ToF-SIMS) have the capability to analysis the nanoparticle surface to describe it in elemental or molecular terms respectively. A recent study reported the possibility of evaluating the surface chemistry of lipid nanoparticles using XPS, necessitating cryogenic conditions, including the detection of PEG on the surface[17]. Extracellular lipid vesicles have also be imaged using ToF-SIMS, however the chemical information obtained from the spectra is not sufficient to identify the different component molecules[18]. While ToF-SIMS depth profiling has previously been employed to characterise the internal structure of giant (10–30 μm) liposomes, it's application toward depth-profiling of lipid particle molecular identification at the nanoscale has not been reported[19]. The development of OrbiSIMS addresses the limitations of ToF-SIMS in analysing high-mass biomolecules with high mass resolving power, whilst utilising gas clusters as an analysis beam provides improved detection of high molecule weight species and preserves molecular information when depth profiling[20].

## Results

A cryogenic preparation protocol was first developed to examine LNPs in vitreous ice sequentially on different areas of the same sample by Cryo-TEM and then Cryo-OrbiSIMS. To test the capability of Cryo-OrbiSIMS depth profiling for the characterisation of the surface and subsurface chemistry of the LNPs, we analysed a model formulation comprising mRNA and 4 lipids; DMG-PEG2k, cholesterol, Dlin-MC3-DMA as the ionisable lipid and DSPC as the helper lipid (Table 1) formed using microfluids. The nanoparticle suspension in phosphate buffered saline (PBS) was deposited onto a plasma etched 10 nm thick amorphous carbon film with regularly spaced 2.4 μm diameter holes supported by copper TEM grids. These were blotted to modulate thickness before plunging into liquid ethane, after which they were transferred under liquid nitrogen for Cryo-TEM imaging (−172 °C). The particles were found by Cryo-TEM to be relatively monodisperse, with their diameters distributed around a mean of 60–80 nm (Fig. 1a) while with DLS they were estimated to be slightly larger with a mean diameter of 91 nm and a PDI of 0.03 (Fig. S1).

Samples were transferred to the OrbiSIMS instrument under liquid nitrogen after TEM imaging for molecular depth profiling under cryogenic conditions (−170 °C). An $Ar_{3000}^+$ gas cluster source was used both as a primary ion analysis and depth profiling beam. Data was acquired from an ensemble of particles simultaneously over a 400 μm × 400 μm area. Since the lateral resolution of the analysis beam is not sufficient to resolve individual LNPs, this approach aimed to maximise the secondary ion count from the LNPs available from such a limited amount of material. Secondary ions diagnostic of lipids were observed in the first spectrum acquired from each sample, consistent with the view of frozen particles supported by the 10 nm thick amorphous carbon support film, rather than particles embedded in ice (Fig. S1). Water clusters from the aqueous suspension are not seen in the OrbiSIMS spectra, a phenomenon common across all frozen hydrated analysis, rationalised as the instability of the water clusters prior to Orbitrap analysis. A ToF-SIMS analysis from the surface of the sample before etching (Fig. S1d) shows water clusters, confirming the hydration of the LNPs, consistent with the model of frozen hydrated LNPs sitting on the carbon support, rather than the embedded alternative shown in Fig. S1c. Secondary ion assignments were guided by the individual component reference spectra (Fig. S2), although in the case of this LNP formulation, all observed fragments were unique to one component and definitively linked to the lipid molecular structures.

### Cryo-OrbiSIMS molecular profiles: DMG-PEG

Since the exterior orientation of the PEG group is well established in the literature for LNPs containing DMG-PEG2k, we first examine the secondary ions from this molecule. Both DMG fragments and sodiated PEG2k fragment ions defined in the standards (Fig. S2) were detected in the SIMS spectra and are plotted as a function of etch time and estimated depth in Fig. 1b (2 further replicates are presented in Fig. S4). The PEG ion fragments in the LNPs are all sodiated and of the form $Na[O-CH_2-CH_2]_x^+$, $Na[O-CH_2-CH_2]_x-O-CH_2^+$ or $NaCH_2[O-CH_2-CH_2]_x-O-CH_2-CH_2^+$. The DMG anchor is detected as the whole molecular fragment: $C_{31}H_{59}O_4^+$, and by ions representing its FA14:0 moiety: $C_{14}H_{27}O_2^+$, $C_{14}H_{27}O_2Na^+$. A complete list is provided in Table S1.

There was a high intensity of PEG signal at the beginning of the analysis which decreased with etching, consistent with the expectation that the PEG portion of the molecule orients outwards for the LNPs. Notably the PEG profile, which is constructed from sodiated PEG ions, does not correlate with the chloride profile derived from the $Na_2Cl^+$ ion (Fig. 1b). This suggests that the sodiated-PEG adduct ions are reflective of an ionic association in the hydrated sample which is an established phenomenon for PEG[21], and that the intensity of these ions is reflective of the amount of PEG in the sample, rather than the $Na_2Cl^+$ distribution. Cationization of PEG through doping prior to SIMS analysis has previously been used to increase the secondary ion yield of oligomers[22]. However, in our experiments the sodium from the PBS particle suspension has enhanced the yield of the PEG from the frozen sample and represents the actual hydrated sample chemistry rather than an artefact of preparation. Notably the PEG in the DMG-PEG2k component spectra in Fig. S2 is also dominated by sodiated secondary ion adducts.

There is near zero intensity of the DMG molecules in the first spectrum acquired, represented in the first data point of the profile indicating that the fatty acid hydrocarbon chains are not present at the outermost surface, fitting the standard description that they are oriented inwards. The PEG and FA14:0 profiles cross at approximately the middle of each ion intensity range at 4–7 nm depth. This measurement can provide an estimate of the thickness of the PEG2k shell. The value is similar to the SANS estimate of a 4 nm thick surface layer of PEG2k assumed to be in a mushroom conformation for similar PEG2k containing particles[5]. A direct comparative analysis of SANS on the same particles would be needed to make a more accurate comparison, but for the purpose of this method development study, the similarity of the value acquired from Cryo-OrbiSIMS depth profiling to that from SANS value reported in previous studies

provides a degree of cautious confidence in the secondary ion profile estimated depth values[5].

The adenine profile is presented to reveal the location of the mRNA molecules in Fig. 1b. Guanine and cytosine are also follow similar profiles as adenine (Figure S3). A 5-methoxyuracil ion, representative of 5-methoxyuridine, was not detected which is most likely due to significantly lower secondary ion formation probability. The 5-methoxyuracil fragment is the only nucleotide without a primary amine group, which is the likely charge centre for the adenine, guanine and cytosine secondary ions.

The adenine ion intensity had near zero intensity at the surface, which increased at a constant rate until ca. 8 nm depth when it plateaued. We interpret the depth at which the adenine signal plateaus to represent the depth at which the core of the particles has been reached. The observation of

adenine intensity before 8 nm is reached, rather than a sharp step up from zero to the value in the core, is interpreted as reflecting a range of PEG overlayer coverage, with the thinner coverage being ablated by sputtering more quickly, whilst thicker areas take longer to sputter through to reach the adenine. This results in the RNA intensity from the core increasing gradually over a range of depths. The DMG ions were equally seen to be revealed gradually, rather than as a step change, consistent with the PEG profile. Since this is an ensemble measurement, this range in PEG coverage may either be within individual particles, or it may represent different particles with different PEG thickness. It is not possible using this approach to determine which of these scenarios is occurring. The molecular orientation findings are summarised in a simplified schematic of the molecular structure for DMG-PEG2k in Fig. 1c to illustrate the interpretation of the

## Table 1 | Individual component structures

| Cholesterol |
| --- |
| Dlin-MC3-DMA |
| DSPC |
| DMG-PEG2K |

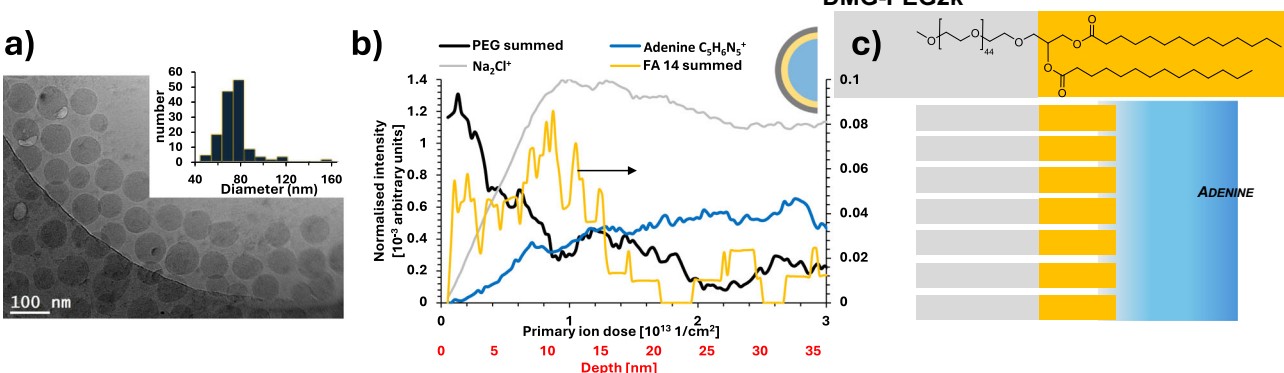

**Fig. 1 | Comparison of the morphology (Cryo-TEM) and chemical composition (Cryo-OrbiSIMS). a** TEM of the LNP formulation with histograms of particle size. **b** Cryo-OrbiSIMS depth profiles with cross sectional schematics of LNP component distribution. The intensity has been normalised to total ion count and the summed FA14:0 ions are presented on secondary intensity scale (indicated by arrow) for the depth profile clarity. The secondary x axis presents depth estimated from comparison with organic standards[30,31]. Further replicates are included in Fig. S4. **c** Simplified schematic of DMG-PEG2k to show the relative molecular orientations.

**Fig. 2 | OrbiSIMS depth profiles of different component fragment ions in Formulation 1.** **a** DSPC, **b** Dlin-MC3-DMA and **c** Cholesterol. The intensity has been normalised to total ion count and the ions: $C_5H_6N_5^+$, $C_6H_{12}NO_2^+$, $C_{37}H_{67}^+$ and $C_{27}H_{45}^+$ are presented on secondary intensity scale (indicated by arrows) for clarity due to the difference in overall peak intensities.

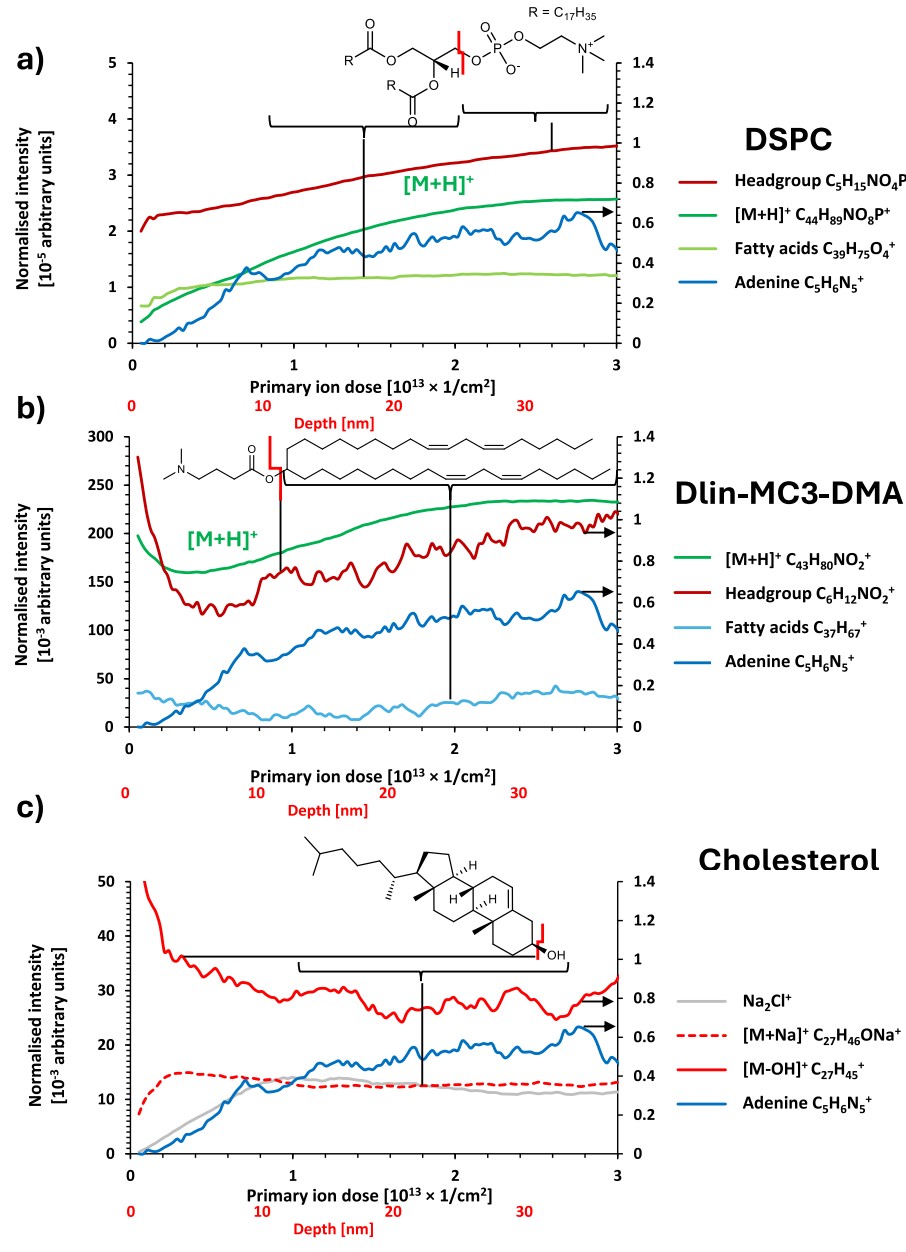

### Ionisable lipid, helper lipid and Cholesterol

In Fig. 2, we present profiles for the DSPC, Dlin-MC3-DMA and Cholesterol lipid components. All DSPC fragments were seen to monotonically increase from the analysis of the outermost surface as profiling was undertaken, although not from zero intensity (Fig. 2a). Adenine representing the RNA core also increased but begins at zero, suggesting that DSPC is present in the shell as well as the core. There is no clear orientation indicated by comparing the headgroup and fatty acid fragments which have similar profiles.

The profile of the ionisable lipid Dlin-MC3-DMA (Fig. 2b) indicated that it was strongly represented at depths less than 3 nm where PEG is also present (<8 nm), thereafter decreasing in intensity and recovering at depths greater than 6 nm. This may represent a propensity for surface segregation, or an association of Dlin-MC3-DMA with the PEG at the exterior, similar to the sodium and chloride ions from the PBS.

Examining the cholesterol ion profiles showed that the positively charged molecular ion after the loss of its hydroxyl group $[M-OH]^+$ was seen

data, but this does not depict the PEG thickness variation inferred from the profile breadth.

to be most intense at the outer surface and decreased within the first 3 nm before reaching a relatively constant value in the interior of the particles (Fig. 2c). This is consistent with cholesterol molecules sitting at the lipid surface with their hydroxyl groups oriented into the hydrophilic region occupied by the PEG, and would be consistent with there being areas with only a thin PEG coverage that allow this $[M-OH]^+$ ion to be present in the earliest analyses.

The intact sodiated molecular ion $[M+Na]^+$ appears to follow a profile which would be expected from an ion generated from the sum of the profiles containing sodium chloride ($Na_2Cl^+$) and cholesterol $[M-OH]^+$, at a low intensity at the outermost surface and increases to a plateau at approximately the same depth of 3 nm. Therefore, we interpret this as an analytical artefact rather than indicating the distribution of a molecular component in the LNP formulation. The cholesterol-Na adduct secondary ion may have formed at the surface during the collision cascade generating secondary ions from both of these species. This is in contrast to the PEG sodium adducts that we did not consider artefactual since their profile that could result from a combination of the profiles of the separate components (the sodium and the PEG).

We note that previous work on pure cholesterol samples prepared without water and without the complicating presence of salts, showed depth profile data of the molecular ion after losing the hydroxyl group $[M-OH]^+$ at the outermost surface and the molecular ion $[M-H]^+$ below it, consistent with an inward molecular orientation of the hydroxyl group[23]. In this LNP system, we do not see an intense $[M-H]^+$ ion, most likely due to adduct formation from sodium present in the PBS. While these two systems are very different and cannot be compared directly, the observation of the different cholesterol fragments suggests molecular orientation in a model pure crystalline system. This is consistent with our observation of molecular orientation of the larger PMG-PEG2k using GCIB etching in this LNP system, suggesting that the orientation of small molecules such as cholesterol may be detectable in some systems.

## Discussion
### RNA-lipid nanoparticles
A state of the art in LNP structural determination method is SANS with contrast-variation to give information on the 3D compositional distribution of different lipids and RNA[5,24,25]. SAXS is used to probe the internal phase of LNPs and the interactions between RNA and ionisable lipids in the LNP core[15]. NMR is also used to provide structural insights[10]. The Cryo-OrbiSIMS depth profiling capability presented here is complementary to these techniques. It is a de novo approach that does not require a pre-conceived model, knowledge of the identity of the lipids or deuterium labelling of components. These attributes would be particularly useful in the analysis of organic nanoparticles comprising more complex component mixtures derived from natural products[26]. This also makes it amenable to the analysis of real-world over model formulations requiring labelling. The analysis throughput can be of the order of 20 depth profiles per hour, making this compatible with high throughput LNP formulation structure-activity investigations and discovery campaigns[27].

Recent advancements in artificial intelligence (AI) that connect lipid chemistry to in vivo performance in high throughput LNP discovery campaigns can be enhanced by incorporating insights on surface structure and composition[28]. These advances in analytics and AI have the potential to facilitate LNP prediction based on formulation composition, which can subsequently inform in vivo efficacy and clinical effectiveness.

Gaining insight into the relative positions of each component within lipid nanoparticles (LNPs) is essential. Such knowledge can help clarify the intricate behaviour of LNPs and contribute to the design of more efficient and safer formulations. Furthermore, it can play a vital role in quality control during the scale-up of manufacturing processes, thereby enhancing the translation of LNPs from the laboratory to clinical applications. Examining the Dlin-MC3-DMA profile suggests that it is located at the surface, with molecular and fragment ions exhibiting a drop in signal intensity from the surface (Fig. 2b). This is mirrored by fragments from all parts of the molecule, headgroup and fatty acid tails. Towards the core, these same ion intensities begin to increase, suggesting that it is also at an elevated amount in the core of the particle. Notably, the depth in the profile where the intensity decreases is at the same position that the fatty acid intensities increase in the DMG PEG molecule, suggesting the presence of a potential hydrophobic barrier layer that could be envisaged as separating molecules like the ionisable lipid from the particle core. Molecules will partition based on their molecular properties, such as charge state and hydrophilicities, as exhibited at the pH of particle formulation in the formation of this hydrophobic barrier layer. As the molecular assembly of LNPs contributes to the determination of LNP structure and thereby function, further experimentation systematically varying the pH of formulation could enable the assessment of this hydrophobic barrier and its impact on molecular distribution.

### Development of Cryo-OrbiSIMS method
The biointerface at the surface of LNPs, comprising the surface lipids identified here and bioadsorbate molecules from the biological milieu in service, is thin and can therefore be regarded as an ideal opportunity for calibration of SIMS for quantification since the matrix effect may be controllable. In model organic systems the matrix effect has been shown to be insignificant for layers of thickness <5 nm[29]. Furthermore, undertaking the analysis in frozen hydrated state is analogous to implantation of oxygen in semiconductors where SIMS is routinely used to quantify dopants, by implanting oxygen to 'normalise' the matrix, providing opportunity for quantification in future[30]. Importantly, the presence of frozen hydrated water also increases the sensitivity to many species compared to dehydrated samples by up to ×10,000[31]. Using standard mixtures to map the relationship between Orbitrap counts, secondary ion yield (counts/nm$^3$)[32,33] and molar molecular concentrations, may allow biointerface molecular quantification, so long as sufficiently large sets of calibration data can be generated and appropriately analysed to produce useful sensitivity factors and to monitor for matrix effects. Control of salts and characterisation of their effects in such calibration datasets will be important since they are unavoidable for real-world samples in vitro and in vivo.

Depth profiles of organic delta layer standards show that the depth accuracy and resolution is dependent on the secondary ions used as well as the primary ions and their energy[34]. Without standard samples comprising molecules from the LNPs, it is difficult to differentiate broadening due to sputter effects from actual concentration gradients representing component intermixing or heterogeneous particle chemistries. Therefore, the depth resolution expected on these spherical, real-world frozen hydrated LNP samples is difficult to estimate, although sputter profiling of nano particulate geometries has been studied for well-defined gold and silica particles where the degradation of the depth content was quantified, this is likely not to transfer directly to the organic self-assembled particles considered here[35]. We have therefore been cautious in our quantitative interpretation of the LNP depth profiles. While standards for depth profile resolution estimation for LNP components are not readily available, supported lipid bi-layers could be envisaged which would have the advantage of being flat, or multi-component micelles, which would well represent the particulate geometry and approximate dimensions. This would allow the variability in PEG overlayer thickness inferred from the broadness of profiles seen herein to be ascribed to actual or experimental depth resolution limits. The new analytical insights this novel method can achieve may be applied to endosomal mimics to examine the influence on LNP components in endosomal escape, offering a powerful framework to improve therapeutic formulations[31,36].

In summary, we find that Cryo-OrbiSIMS depth profiling offers a valuable methodology for label free characterisation of the native LNP surface molecular structure and stratification in the frozen hydrated state. In addition to this high-fidelity insight into the surface molecular stratification of the individual components, the orientation of the PEG-lipid can be identified. This orientation is consistent with models from neutron scattering and well-established descriptions of similar LNPs. It is anticipated that this approach will find application in the rapidly developing RNA-LNP therapeutics space. It will also be useful in organic biomaterials more broadly, where capturing native hydrated surface chemistry and identifying molecular constituents and their orientation, are critical to engineer their performance.

## Methods
### LNP preparation
LNPs were formulated via microfluidic mixing using a Precision Nanosystems NanoAssemblr Ignite instrument with a formulation were composed of Dlin-MC3-DMA/DSPC/DMG-PEG2k/Cholesterol at a molar ratio of (50/10/1.5/38.5) with an N:P ratio of 6. Lipids were solubilized in ethanol. The RNA cargo was comprised of a 1:1 mol:mol of the following constructs: CleanCap® Firefly Luciferase mRNA (5-methoxyuridine) and CleanCap® Erythropoietin mRNA (5-methoxyuridine) formulated in a 50 mM citrate buffer at pH 3.0. After mixing, nanoparticles were transferred to Slide-A-lyzer G2 dialysis cassettes (10 k MWCO) and dialysed into PBS in at least 200x excess buffer for two hours at room temperature followed by overnight dialysis with gentle

stirring in refreshed buffer at 4 °C. LNP size and polydispersity were determined using a Zetasizer Ultra Malvern Panalytical instrument and are presented in Fig. 1a.

LNP encapsulation efficiencies were evaluated using the Invitrogen™ Quant-it™ RiboGreen Reagent and RNA Assay Kit (Thermo Fisher).

## Cryo-TEM

Quantifoil R2/1 Cu 300 TEM grids (Agar Scientific Ltd, Essex UK) were glow discharged prior to use to render them hydrophilic. LNP suspensions were automatically blotted with contact between the TEM grid surface and filter paper was pre-set to 3 or 6 s. They were then frozen in vitrified ice on these grids in liquid ethane using a Leica EM GP2 automatic plunge freezer and maintained under liquid nitrogen until required for analysis. A blotting time of 6 s yielded satisfactory results for samples the 4 mg/mL concentration samples presented in this paper.

Samples were maintained at cryogenic temperatures during transfer and analysis using an Elsa™ cryo-transfer holder and temperature controller (Gatan). Imaging was performed on a JEOL 2100 F FEG-TEM operating at 200 kV and recorded on a K3 direct electron detection camera (Gatan), nominal defocus was applied and an objective aperture inserted to improve image quality. A sample temperature of −172 °C (±0.5 °C) was maintained during imaging. Images in this document were binned and converted to JPEG format to improve clarity, raw data is available online.

## Cryo-OrbiSIMS analysis

The samples post Cryo-TEM were retained in frozen hydrated state and transferred into the Leica VCM bath filled with liquid nitrogen. The samples were fixed to the Leica cryogenic block and introduced into the airlock on the cryo-sample holder via the Leica vacuum transfer system. The analysis was carried out at −170°C using a closed-loop liquid nitrogen pumping system (IONTOF GmbH).

For the acquisition of the OrbiSIMS depth profiles, a 20 keV $Ar_{3000}^+$ analysis beam of 20 μm diameter, was used as primary ion beam. The distribution is shown in Fig. S5. OrbiSIMS profiles were acquired from 3 replicates from the same LNP batch. $Ar_{3000}^+$ with duty cycle set to 4.2% and GCIB current was 220 pA. The Q-Exactive depth profile was run on the area of 400 μm × 400 μm using random raster mode with crater size 498 × 498 μm. The cycle time was set to 400 μs. Optimal target potential was set to −57 V in negative polarity and +57 V in positive polarity. Argon gas flooding was in operation in order to aid charge compensation, pressure in the main chamber was maintained at $9.0 \times 10^{-7}$ mbar. The spectra were collected in positive polarity, in mass range $m/z$ 75–1125. The injection time was set to 500 ms and each area analysed lasted 1000 scans, the total ion dose per measurement was $2.62 \times 10^{14}$ ions/cm². The static limit of primary ion dose, commonly used for primary beams which result in the build-up of sample damage ($10^{12}$ ions/cm²) is reached at the third scan. Mass-resolving power was set to 240,000 at $m/z$ 200. Etching removes the particle strata which are sampled sequentially as primary ion bombardment proceeds, with the exception of shadowing and preferential sputter yields discussed in the Yang et al.[35].

Calibration of the Orbitrap analyser was performed on the silver sample, using silver clusters following the method described by Passarelli et al. using the $Bi_1^+$ liquid metal ion gun as a primary ion beam[20].

We have used information from a published depth profiling organic molecular standard (Irganox) to estimate the depth from the primary ion dose, which is displayed as a secondary x axis (red) in Fig. 1b[37]. The sputter yield from 20 keV $Ar_{3000}^+$ on Irganox was corrected for the reduced temperature (−170 °C) used in this work compared to the literature values[38] using the calculations developed by Seah et al.

Each depth profile was acquired from the full thickness of the sample. An initial nanoparticle components region was first seen, followed by a complex mixture of inorganic ions (not shown) interpreted to be from the surface of the carbon support substrate were found, appearing in the profile below the particles before the substrate was reached. We do not interpret the relative positions of these ions, instead focussing on the organics which represent the lipids.

While depth profiling is commonly used in SIMS for the analysis of layered systems[39], it is limited to assessing depth distributions below half of the particle diameter due to the cumulative development of topography in the etching process[40]. Therefore the data in this paper that is interpreted is limited to a primary ion dose of $3 \times 10^{13}$ ions/cm².

## Time of flight secondary ion mass spectrometry (ToF SIMS)

For the acquisition of liquid metal ion gun (LMIG) ToF-SIMS spectra of the lipid nanoparticle formulation, a 30 keV $Bi_3^+$ primary beam was used. LMIG current was 0.03 pA. The acquisition was run on the area of 500 μm × 500 μm using random raster mode. Optimal target potential was set to +58 V. The total ion dose per measurement was $7.36 \times 10^9$. The analysis was carried out at −170 °C using a closed-loop liquid nitrogen pumping system (IONTOF GmbH).

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

## Acknowledgements
The OrbiSIMS facility was funded by the EPSRC Strategic Equipment Grant '3D OrbiSIMS: Label free chemical imaging of materials, cells and tissues' EP/P029868/1. The CryoTEM holder and detector were funded by the EPSRC Research Grant 'A New Correlative Approach for Structure Determination & Imaging of Molecular Materials' EP/W006413/1. Reece Franklin of University of Nottingham assisted with cryogenic sample preparation of replicate analyses. Kerry Benenato of Sail Biomedicines read and commented on the manuscript during its drafting and Aditi Borkar is acknowledged for providing RNA sequence advice in the response to referees.

## Author contributions
A.M.K. acquired and processed the Cryo OrbiSIMS, read and commented on the manuscript. M.F. acquired TEM images and read and commented on the manuscript. J.A.W. prepared the TEM/Cryo OrbiSIMS samples for analysis and quantified the TEM particle images. I.S.G. advised on SIMS data processing and interpretation and read and commented on the manuscript. D.J.S. advised on SIMS sample preparation, data processing and interpretation and read and commented on the manuscript. A.H., V.C., C.E.P., D.D., S.P., M.U., and M.R.A. designed formulations to evaluate, helped guide data analysis and interpretation, and read and commented on the manuscript. A.H. and S.P. undertook Nanoparticle synthesis and characterisation. R.L. advised on undertaking the study and read and commented on the manuscript. M.R.A. led writing of the manuscript and response to referees, and supervised data acquisition, processing and interpretation.

## Competing interests
M.R.A., A.M.K., M.F., J.A.W., and D.J.S. at the University of Nottingham have carried out paid work for Senda Biosciences (now Sail Biomedicines). Sail Biomedicines is a for-profit organization that aims to translate circular RNA technology and proprietary programmable nanoparticles, utilising natural components, to programme medicines. No proprietary systems are included in this manuscript. A.H., C.E.P., V.C., S.P., D.D., and MU are employed by and have equity interests in Sail Biomedicines. For a list of entities with which R.L. is, or has been recently involved, compensated or uncompensated, a live copy can be accessed at https://www.dropbox.com/scl/fi/xjq5dbrj8pufx53035zdf/RL-COI-2024.pdf?rlkey=fwv336uoepiaiyg4e7jz5t4zo&dl=0. The remaining author declares no competing interests.
