## [Transparent Peer Review file · Communications Chemistry]

Study on molecular orientation and stratification in RNA-lipid nanoparticles by Cryogenic Orbitrap Secondary Ion Mass Spectrometry

Corresponding Author: Professor Morgan Alexander

Version 0:

Reviewer comments:

Reviewer #1

(Remarks to the Author)

This manuscript uses a newly developed cryo OrbiSIMS method to examine the molecular orientation and stratification of PEG, fatty acid (FA), and RNA components in lipid nanoparticles (NPs) similar to those used for mRNA drug delivery platforms. The identity of the lipids that present at the NP surface play a role in how they interact with and are perceived by the body and their resultant potency. They use a model formulation to develop cryogenic sample preparation for molecular depth profiling Orbitrap secondary ion mass spectrometry (Cryo-OrbiSIMS) and profiling PEG and FA14:0 they provide an estimate of the thickness of the PEG2k nanoparticle shell and adenine profiling indicates the core of the nanoparticle. This is a really nice result. This manuscript is really a demonstration of a method.

The real advantages to what is shown here are, 1) The OrbiSIMS approach is a de novo approach that does not require a preconceived model, knowledge of the identity of the lipids or deuterium labelling of components, or analysis of real-world rather than labelled model formulations, and 2) The analysis throughput can be of the order of 20 profiles per hour, making this compatible with high throughput LNP formulation structure-activity investigations and discovery campaigns.

Overall, the conclusions are justified by the data, but there are a few places this might be discussed more. I would suggest a minor revision before publishing.

Comments:

The authors state, "molecular depth profiling of LNPs is possible using cryogenic OrbiTrap TM secondary ion mass spectrometry (Cryo-OrbiSIMS)." And "found that the depth distribution of individual lipid components is revealed relative to the surface and the RNA cargo defining the core." It would be nice if the authors stated the overall importance of this finding. Do the structural details they find have implications on nanoparticle behaviour? Do they change the critical stage of endosomal escape in transfection, for which efficiencies are estimated to be as low as 2%. Can this be clarified?

The graphical abstract is complicated and still does not get across the critical message of the manuscript.

When cryo TEM is used this should be indicated. The authors write in the Results, "Samples were transferred to the OrbiSIMS instrument under liquid nitrogen after TEM imaging for molecular depth profiling under cryogenic conditions (-170°C)." making it seem they used cryo TEM, but it is not clearly stated.

It does not seem that DLS and PDI are defined.

The authors write that the guanine and cytosine profile across the nanoparticle are similar. Do they mean to adenine and is it possible to perhaps include these data to be more convincing?

Missing an "and" in the sentence, "While these two systems are very different cannot to be compared directly,"

Figure 2b has five ions listed, but only four colors and lines in the plot. Furthermore, the data are only really convincing at depth to about 5 nm. The authors acknowledge this but it might be made a little more clear as the deeper signals should be the adenine showing the RNA core and this is discussed a great deal in the manuscript.

The paragraph on AI seems out of place and unconnected to the rest of the manuscript.

The Discussion can be shortened. Some of this is rehash of the Introduction and some seems unrelated and should either

be put in context to the work presented here or omitted.

The Summary is poorly written with grammatical and punctuation errors and seems almost rushed. It is also not a very clear statement of what is in this manuscript.

Is Figure S1a really just a straight line?

Figure S3, can the authors indicate why uracil is not observed in the SIMS spectra?

Reviewer #2

(Remarks to the Author)

The manuscript of Kotowska et al. presents a new methodology for the spatially resolved characterisation of lipid nanoparticles. Lipid nanoparticles are very important i.e. as carrier for drugs. Usually the local structure of the nanoparticles is obtained by neutron scattering. Here the authors use high resolution mass spectrometry (OrbiSIMS) for the elucidation of the composition from core to shell. This approach is highly innovative and as the presented results provide important and precise information about the 3D composition. It is worth to be reported in "Nature Communications in Chemistry". However I have three major points that should be clarified / improved before publication.

- 1) Within the manuscript no information, obtained by established methods, about the structure of the nanoparticles is given. For such work, I would expect that an established method like SANS is used to initially characterize and later discuss the reliability and added value of the new method. The shown TEM data give no hint to the lipid structure.
- 2) It is not clear whether the analysis was repeated on several samples. Please state clearly in the experimental section how many repetitions were carried out.
- 3) The nanoparticles have a spherical structure. Therefore SIMS depth profiling data origins from different height levels and in the depth profile information from different areas is mixed. Please discuss / comment on the effect of the geometry with respect to the analysis.

I have also a few minor comments:

- 1) The arrows with the text inside are not so nice for a graphical abstract. Maybe the SIMS process can be better illustrated by a illustration with a beam and "flying ions".
- 2) L. 126: The abbreviation "PDI" is not introduced.
- 3) L.188/189: Something is missing in the first part of this sentence. What cannot be determined?
- 4) Fig. 1b: It is not clear for the reader which signal intensities are belonging to the second intensity axis.
- 5) Fig. 2: It is really hard to correlate which masses belongs to which intensity scale. Maybe this information can be added to the graphs with arrows?
- 6) It is quite unusual that a pure Ar3000+ beam is produced with such a high current. Usually, in the IONTOF GCIB sources a beam with a cluster distribution is used. Please comment on this and give the cluster distribution in SI.
- 7) L. 381. The given area should have the dimension of an area (μm^2 instead of μm). Please check also the following text.
- 8) L. 385: I doubt that an analysis is possible with such a high Ar pressure in the main chamber. Please check the dimension.

Reviewer #3

(Remarks to the Author)

The manuscript presents a novel methodology to characterize the spatial distribution of different molecular components in RNA-containing lipid nanoparticles (LNPs) by OrbiSIMS analysis under cryogenic sample temperatures. The method relies on retention of the LNPs in their native hydrated state in ice using cryo-preparation methods adopted from cryo-TEM, and the structural integrity of the LNPs is also verified by cryo-TEM prior to the cryo-OrbiSIMS analysis. The method should be of very high interest to the LNP research field, as it can potentially provide important information for the performance of the LNPs (i.e., spatial distributions of individual lipid and RNA components in the LNPs) that is otherwise difficult to obtain. A major claim is that the method shows molecular stratification of individual components, e.g., the PEG-lipid, for which the PEG chain is shown to be oriented outwards on the LNP surface, whereas the lipid part is localised more to the inner part of the LNP, in agreement with previous models. The paper is, in general, clearly written and well organized, although multiple errors and some unclear formulations need to be corrected by the authors, see below. The results are convincing and the conclusions are well founded. However, my opinion is that the authors need to consider the following points in order to improve the quality and impact of the paper.

My major concern relates to the hydrated state of the LNPs during the OrbiSIMS analysis.

The authors state that the method relies on the retention of the LNPs in the hydrated state. However, I would like to see evidence that this is actually the case during the OrbiSIMS analysis. The TEM image acquired prior to SIMS analysis indeed verifies structural integrity, but can the authors verify that the sample is hydrated also after transfer into the OrbiSIMS instrument? If the sample is hydrated (i.e., embedded in ice), I would expect to see data for ions associated with water, such as $(\text{H}_2\text{O})_n\text{H}^+$ and/or $(\text{H}_2\text{O})_n\text{Na}^+$ cluster ions. Can the authors show depth profiles for such water-associated ions? Additionally, the use of Na_2Cl^+ to represent Na in the buffer is not very convincing, if we assume that the LNPs are embedded in ice. Instead, ions of the type $(\text{H}_2\text{O})_n\text{Na}^+$ would be a better choice. Can the authors provide depth profiles for

ions that more realistically can represent buffer-associated Na?

Finally, to explain why the adenine depth profile shows a gradual increase and not a sharp step at the interior of the LNP, the authors argue that an inhomogeneous thickness of the PEG layer on the LNP surface would lead to such a gradual increase. This is correct if PEG is the only component on the LNP surface, i.e., if the LNPs are not embedded in ice. If embedded in ice, the sputter time needed to reach the mRNA would not be greatly affected by the thickness of the PEG layer. Other possible explanations for the gradual increase could be the spherical topography of the LNPs and/or inhomogeneous spatial distributions between different LNPs.

So, the question is how the authors envision the hydrated state. Are the LNPs embedded in a thin film of ice (as typically envisioned for the cryo-TEM preparation used in this work) or has the bulk ice disappeared, with the structural integrity of the LNPs stabilized by the low temperature and/or individual water molecules directly bound to the lipid structures. Without evidence for bulk ice in the sample, the latter possibility seems more likely to me. A more detailed discussion about the "hydrated state" would definitely be beneficial and clarifying for how to interpret the data. Additionally, transfer back to the TEM and cryo-TEM analysis after cryo-OrbiSIMS would provide conclusive evidence for the structural integrity of the LNPs during the cryo-OrbiSIMS analysis.

Detailed comments:

The manuscript contains numerous obvious errors and some unclear formulations that need attention by the authors, some of which are listed below. In addition, I recommend the authors to make a detailed review of the entire manuscript (text and figures) to find and correct possible additional errors.

Page 3, "The also enable...": A word seems to be missing

Page 4 "...to directly analysis the nanoparticle..." Should be "...analyse...?"

Page 5 "(Figure S1)", two places Should be "(Figure S2)"?

Page 5 The $C_xH_{2x}O_{x/2+1}$ etc formulas need to be checked, do not agree with the mentioned examples, also in Figure S2

Page 5 "A complete list is provided in Table S1" The list is not complete, contains only the PEG fragments. Change text to "A list of the PEG fragments..." or add the FA14:0 ions to the list.

Page 6 "...levels off to approximate to the core of the particles..." Unclear

Page 6 "...in the first analysis of the profile..." Unclear, change "analysis" to "data point", or similar

Page 6 "...results in a the RNA intensity..." Remove "a"?

Page 7 "...which, although extending this beyond this initial proof-of-concept to more LNP formulations, the PEG thickness variability will likely be able to be correlated anticipated LNP structure, e.g. by varying the PEGylated lipid amount." I don't understand the meaning of this, reformulate.

Page 7, Figure 1 caption References 31, 33 seems incorrect to me, check.

Page 8 Last sentence starting "While these two systems..." contains at least two errors and is otherwise also very difficult to decipher. Correct and reformulate.

Page 9, Figure 2b The labels include four trace colors but five assignments, one assignment must be removed

Page 12 "...is critical engineering their performance." A word is missing?

Page 13 RiboGreen RNA assay kit Add manufacturer/reference

Page 15 The entire paragraph starting "Each depth profile was acquired..." does not make sense to me, do not understand. Reformulate.

Page 18, Figure S2 "C54H89NPO8+" Should be "C44H89NPO8+"?

Page 22 "Table 1" Should be "Table S1"?

Version 1:

Reviewer comments:

Reviewer #1

(Remarks to the Author)

Dear Authors,

Thank you for answering my questions related to the manuscript. I have a few comments on the revised version of the manuscript. Please find the comments, and the manuscript with highlighted parts related to my comments.

Reviewer #2

(Remarks to the Author)

The authors have carried out the revision carefully. All open questions have been answered satisfactorily. So, I recommend the manuscript for publication.

Reviewer #3

(Remarks to the Author)

I am satisfied with the changes made in the manuscript by the authors in response to my comments. For me, the manuscript is now acceptable for publication.

Response to Chemical Communications reviewers' comments

We thank the reviewers for the time and close attention they have given to the manuscript with their expertise. It will certainly improve it. We have made every effort within our capability to address the points made by the reviewers below (in green text following their comments), and with tracked modifications to the submitted manuscript.

Reviewers' comments:

Reviewer #1

It would be nice if the authors stated the overall importance of this finding. Do the structural details they find have implications on nanoparticle behaviour? Do they change the critical stage of endosomal escape in transfection, for which efficiencies are estimated to be as low as 2%. Can this be clarified?

This paper has a novel method development focus. We would not want to overstretch the interpretation of our data to the complex environment of endosomal release and its poor efficacy based upon one LNP composition. We have clarified this at the end of the discussion:

The new analytical insights this novel method can achieve can be paired with endosomal mimics to examine the influence on LNP components in endosomal escape, offering a powerful framework for to improve therapeutic formulations.³⁴

The graphical abstract is complicated and still does not get across the critical message of the manuscript.

When cryo TEM is used this should be indicated. The authors write in the Results, "Samples were transferred to the OrbiSIMS instrument under liquid nitrogen after TEM imaging for molecular depth profiling under cryogenic conditions (-170°C)." making it seem they used cryo TEM, but it is not clearly stated.

Thank you-corrected.

It does not seem that DLS and PDI are defined.

Thank you-corrected.

The authors write that the guanine and cytosine profile across the nanoparticle are similar. Do they mean to adenine and is it possible to perhaps include these data to be more convincing?

Yes, similar to adenine which has been clarified. These data are all in Figure S3, the indication of which has been moved forward in the sentence to make it clearer.

Missing an "and" in the sentence, "While these two systems are very different cannot to be compared directly,"

Thank you-corrected.

Figure 2b has five ions listed, but only four colors and lines in the plot.

Thank you-corrected.

Furthermore, the data are only really convincing at depth to about 5 nm. The authors acknowledge this but it might be made a little more clear as the deeper signals should be the adenine showing the RNA core and this is discussed a great deal in the manuscript.

The authors feel that it is important to clarify that the thickness below which the matrix effect is minimised is 5 nm. We discuss this in the last paragraph on page 11 regarding the limitations of the depth estimates in this early stage of such a novel approach. The depth resolution will degrade with depth, but for this sample format this remains to be quantified. However, we do have confidence that the adenine and other RNA signals are reliably generated from the LNP core.

The paragraph on AI seems out of place and unconnected to the rest of the manuscript.

We would argue that the paragraph is very relevant to the role this analysis technique will likely assume given the role of AI in LNP discovery as well as other biomaterials. We placed it directly after the section describing recent high throughput advances, but have reduced it and tried to signpost it better in this resubmitted manuscript.

The Discussion can be shortened. Some of this is rehash of the Introduction and some seems unrelated and should either be put in context to the work presented here or omitted.

We have considered all specific points by the reviewers and made changes in the discussion noted throughout. Given this is the first sight of this data type we think a significant discussion of the data here and how this method needs to develop is warranted. If the reviewer has some more specific requests we can address these, as the AI one above where a reduction was made.

The Summary is poorly written with grammatical and punctuation errors and seems almost rushed. It is also not a very clear statement of what is in this manuscript.

This has been modified, but retains the aim of being a high level description of the method and its potential future application to complement the details that has been presented previously.

Is Figure S1a really just a straight line?

Figure S1a is a table. We have modified it to clarify this.

Figure S3, can the authors indicate why uracil is not observed in the SIMS spectra?

As noted on page 6 this is “most likely due to significantly lower secondary ion formation probability”.

Reviewer #2 (Remarks to the Author):

The manuscript of Kotowska et al. presents a new methodology for the spatially resolved characterisation of lipid nanoparticles. Lipid nanoparticles are very important i.e. as carrier for drugs. Usually the local structure of the nanoparticles is obtained by neutron scattering. Here the authors use high resolution mass spectrometry (OrbiSIMS) for the elucidation of the composition from core to shell. This approach is highly innovative and as the presented results provide important and precise information about the 3D composition. It is worth to be reported in "Nature Communications in Chemistry". However I have three major points that should be clarified / improved before publication.

1) Within the manuscript no information, obtained by established methods, about the structure of the nanoparticles is given. For such work, I would expect that an established method like SANS is used to initially characterize and later discuss the reliability and added value of the new method. The shown TEM data give no hint to the lipid structure.

We have specifically chosen a well-researched LNP system, for which the DMG-PEG structure is universally agreed upon in the literature. In no publications is the external orientation of PEG in the DMG-PEG disputed-indeed it is required to produce and stabilise monodisperse LNPs. It is this orientation that we use to validate our observation with the novel Cryo-OrbiSIMS technique. The PEG thickness was quantified in Arteta et al 2018 using SANS, although this is not required to validate our main finding, i.e. we can detect molecular orientation using Cryo-OrbiSIMS depth profiling. Undertaking small angle neutron scattering (SANS) analysis is a very major undertaking, not least the need to apply for and schedule beam time, and we do not regard that replicating the well regarded Arteta paper in this paper would be beneficial to our main claim.

Our LNP production method was validated morphologically using cryo-TEM and DLS. The observation that the particles are physically as to be expected gives us confidence that the LNP structure is as reported in the literature. This is consistent with our OrbiSIMS findings, which for the DMG-PEG orientation are very clear.

2) It is not clear whether the analysis was repeated on several samples. Please state clearly in the experimental section how many repetitions were carried out.

The number of replicates was noted in the results section, and has now been duplicated in the methods section to satisfy this request.

3) The nanoparticles have a spherical structure. Therefore SIMS depth profiling data origins from different height levels and in the depth profile information from different areas is mixed. Please discuss / comment on the effect of the geometry with respect to the analysis.

The effect of particle geometry on SIMS is expanded upon briefly on page 11 in response to this request. Briefly, the height of the particle is not of importance, rather that the particles' strata are sampled sequentially as etching proceeds. The exception of shadowing and preferential sputter yields discussed in the Yang et al.

paper is now cited along with the inclusion of an explanation of this concept in the methods section:

Etching removes the particle strata which are sampled sequentially as primary ion bombardment proceeds, with the exception of shadowing and preferential sputter yields discussed in the Yang et al.³⁹

I have also a few minor comments:

1) The arrows with the text inside are not so nice for a graphical abstract. Maybe the SIMS process can be better illustrated by a illustration with a beam and "flying ions".

We agree and have modified this.

2) L. 126: The abbreviation "PDI" is not introduced.

Thank you-corrected.

3) L.188/189: Something is missing in the first part of this sentence. What cannot be determined?

We presume the reviewer means L243 where the word and was omitted: '...are very different and cannot be compared...' which has been corrected with the addition of the word "and".

4) Fig. 1b: It is not clear for the reader which signal intensities are belonging to the second intensity axis.

This is noted in the caption-and is now also clarified in the graph itself using arrows for those on the secondary axis.

5) Fig. 2: It is really hard to corelate which masses belongs to which intensity scale. Maybe this information can be added to the graphs with arrows?

This is noted in the caption-and is now also clarified in the graph itself for those on the secondary axis.

6) It is quite unusual that a pure Ar3000+ beam is produced with such a high current. Ususally, in the IONTOF GCIB sources a beam with a cluster distribution is used. Please comment on this and give the cluster distribution in SI.

This is correct, the GCIB beam is not pure Ar 3000+ but is normally abbreviated as such. The cluster distribution is now noted in Figure S5.

7) L. 381. The given area should have the dimension of an area (μm^2 instead of μm). Please check also the following text.

This has been corrected to $400 \mu\text{m} \times 400 \mu\text{m}$ rather than μm^2 to avoid the potential confusion between $400 \mu\text{m} \times 400 \mu\text{m}$, $400 \mu\text{m}^2$ and $160,000 \mu\text{m}^2$

8) L. 385: I doubt that an analysis is possible with such a high Ar pressure in the main chamber. Please check the dimension.

Argon gas flooding was in operation in order to aid charge compensation, with pressure in the main chamber maintained at 9.0×10^{-7} mbar.

Reviewer #3 (Remarks to the Author):

The results are convincing and the conclusions are well founded. However, my opinion is that the authors need to consider the following points in order to improve the quality and impact of the paper.

My major concern relates to the hydrated state of the LNPs during the OrbiSIMS analysis.

The authors state that the method relies on the retention of the LNPs in the hydrated state. However, I would like to see evidence that this is actually the case during the OrbiSIMS analysis. The TEM image acquired prior to SIMS analysis indeed verifies structural integrity, but can the authors verify that the sample is hydrated also after transfer into the OrbiSIMS instrument? If the sample is hydrated (i.e., embedded in ice), I would expect to see data for ions associated with water, such as $(\text{H}_2\text{O})_n\text{H}^+$ and/or $(\text{H}_2\text{O})_n\text{Na}^+$ cluster ions. Can the authors show depth profiles for such water-associated ions?

1. The vision of “embedding” of the particles is understandable given the frozen hydrated nature of the sample, but is in fact a misconception. The below

situation is instead expected due to optimisation of the blotting time, the surface tension of the water and the geometry and relative dimension of the particles and grid components:-

Embedding (ice overlayer):

Supported frozen particles incorporating water:

Figure S1c) Frozen nanoparticle schematic for the Cryo-OrbiSIMS analysis..

This arrangement is consistent with there being LNPs represented in the first spectra acquired before etching commences. If this was pure ice, they would not be present and would be observed later in the depth profile. We propose including this in the SI, with the description:

“Lipid secondary ions are seen in the first spectrum acquired from each sample, before etching commenced, consistent with the view of frozen particles supported by the 10 nm thick amorphous carbon support film, rather than particles embedded in ice (Figure S1).”

The absolute intensities of the lipid nanoparticle components at the first scan of the analysis are:

Peak assignment	Component name	Intensity [a.u.]
Summed ions	FA 14:0 summed	0
Summed ions	PEG summed	7436.56
C ₂₇ H ₄₅ ⁺	cholesterol	10023.16
C ₂₇ H ₄₆ ONa ⁺	cholesterol	37215.19
C ₄₃ H ₈₀ NO ₂ ⁺	DLin-MC3-DMA	984544.03
C ₄₄ H ₈₉ NO ₈ P ⁺	DSPC	21577.38
C ₅ H ₆ N ₅ ⁺	RNA adenine	0

2. The Orbitrap detector with SIMS does not include water ions as water clusters in the same way as in ToF SIMS. This is likely due to the delicate water complexes falling apart whilst awaiting injection into the Orbitrap, or during the argon cell used to reduce the ion energy to allow their acceptance by the OrbiTrap analyser. The molecular water ion, OH₂⁺ appears at a mass below the minimum for inclusion into the Orbitrap spectrum. However, we do have ToF SIMS analysis from additional experiments that we did not previously include, giving us confidence that the samples are frozen. This has been inserted into the SI as below:

Figure S1 ...d) LMIG ToF surface analysis spectra of the LNP formulation acquired in cryogenic conditions. The repetitive pattern represents water cluster peaks, the first 3 water cluster peaks are assigned in the spectrum, with the other indicated by asterisk.

Additionally, the use of Na_2Cl^+ to represent Na in the buffer is not very convincing, if we assume that the LNPs are embedded in ice. Instead, ions of the type $(\text{H}_2\text{O})_n\text{Na}^+$ would be a better choice. Can the authors provide depth profiles for ions that more realistically can represent buffer-associated Na?

The buffer is the only component that includes significant Na_2Cl^+ intensity as can be seen in Figure S2. There are no water- Na^+ ions in the ToF spectrum now presented in Figure S1d. The Na_2Cl^+ ion is not normally considered to be controversial as an indicator of sodium chloride for those used to dealing with Orbitrap data, but we have inserted a clarification on the point of water clusters.

“Water clusters from the aqueous suspension are not seen in the OrbiSIMS spectra, a phenomena common across all frozen hydrated work, rationalised as the instability of the water clusters prior to Orbitrap analysis.”

Finally, to explain why the adenine depth profile shows a gradual increase and not a sharp step at the interior of the LNP, the authors argue that an inhomogenous thickness of the PEG layer on the LNP surface would lead to such a gradual increase. This is correct if PEG is the only component on the LNP surface, i.e., if the LNPs are not embedded in ice. If embedded in ice, the sputter time needed to reach the mRNA would not be greatly affected by the thickness of the PEG layer. Other possible explanations for the gradual increase could be the spherical topography of the LNPs and/or inhomogeneous spatial distributions between different LNPs. So, the question is how the authors envision the hydrated state. Are the LNPs embedded in a thin film of ice (as typically envisioned for the cryo-TEM preparation used in this work) or has the bulk ice disappeared, with the structural integrity of the LNPs stabilized by the low temperature and/or individual water molecules directly bound to the lipid structures. Without evidence for bulk ice in the sample, the latter possibility seems more likely to me. A more detailed discussion about the “hydrated state” would definitely be beneficial and clarifying for how to interpret the data. Additionally, transfer back to the TEM and cryo-TEM analysis after cryo-OrbiSIMS would provide conclusive evidence for the structural integrity of the LNPs during the cryo-OrbiSIMS analysis.

As noted above, LNP components appear in the first spectra. Hopefully this is now clear to the reviewer with the addition of the ToF SIMS and the explanation of the geometry in Figure S1c.

Detailed comments:

The manuscript contains numerous obvious errors and some unclear formulations that need attention by the authors, some of which are listed below. In addition, I recommend the authors to make a detailed review of the entire manuscript (text and figures) to find and correct possible additional errors.

We thank the reviewer for pointing out these typos, they have been attended to, and the manuscripts has been additionally proofread.

Page 3, “The also enable...”: A word seems to be missing

Page 4 “...to directly analysis the nanoparticle...” Should be “...analyse...”?

Page 5 “(Figure S1)”, two places Should be “(Figure S2)”?
Page 5 The $CxH2xOx/2+1$ etc formulas need to be checked, do not agree with the mentioned examples, also in Figure S2

This has been corrected and the table s1 expanded to clarify the structural assignment further.

Page 5 “A complete list is provided in Table S1” The list is not complete, contains only the PEG fragments. Change text to “A list of the PEG fragments...” or add the FA14:0 ions to the list.

The caption stipulated that the DMG FA fragments were not included. We have not added the FA(14:0) fragments to the list.

Page 6 “...levels off to approximate to the core of the particles...” Unclear

Clarified thus: “interpret the depth at which the adenine signal levels off to represent the depth at which the core of the particles has been reached.”

Page 6 “...in the first analysis of the profile...” Unclear, change “analysis” to “data point”, or similar

Clarified to: “There is near zero intensity of the DMG molecules in the first spectrum acquired, represented in the first data point of the profile”

Page 6 “...results in a the RNA intensity...” Remove “a”?

Done.

Page 7 “...which, although extending this beyond this initial proof-of-concept to more LNP formulations, the PEG thickness variability will likely be able to be correlated anticipated LNP structure, e.g. by varying the PEGylated lipid amount.” I don’t understand the meaning of this, reformulate.

To clarify we have expanded to “It is not possible using this approach to determined which of these alternative scenarios is occurring, beyond this initial proof-of-concept, the PEG thickness variability will likely be able to be correlated and anticipated from varying the LNP formulation, e.g. by modifying the PEGylated lipid amount.”

Page 7, Figure 1 caption References 31, 33 seems incorrect to me, check.

Thank you-these are now re-joined to the reference manager and display the correct citation.

Page 8 Last sentence starting “While these two systems...” contains at least two errors and is otherwise also very difficult to decipher. Correct and reformulate.

Thank you. Sentence split into 2 to make clearer: “While these two systems are very different and cannot to be compared directly, the observation of the different cholesterol fragments suggests molecular orientation in a model pure crystalline system. This is consistent with our observation of molecular orientation of the larger PMG-PEG2k using

GCIB etching in this LNP system, suggesting that the orientation of small molecules such as cholesterol may be detectable in some systems.”

Page 9, Figure 2b The labels include four trace colors but five assignments, one assignment must be removed

This has been corrected.

Page 12 “...is critical engineering their performance.” A word is missing?

Thank you: “is critical for engineering their performance”

Page 13 RiboGreen RNA assay kit Add manufacturer/reference

This clarification has been inserted: “Invitrogen™ Quant-it™ RiboGreen Reagent and RNA Assay Kit (Thermo Fisher)”

Page 15 The entire paragraph starting “Each depth profile was acquired...” does not make sense to me, do not understand. Reformulate.

Reformulated to be clearer:

Each depth profile was acquired from the full thickness of the sample. An initial nanoparticle components region was first seen, followed by a complex mixture of inorganic ions (not shown) interpreted to be from the surface of the carbon support substrate were found, appearing in the profile below the particles before the substrate was reached. We do not interpret the relative positions of these ions, instead focussing on the organics which represent the lipids.

Page 18, Figure S2 “C54H89NPO8+” Should be “C44H89NPO8+”?

Thank you – corrected.

Page 22 “Table 1” Should be “Table S1”?

We have modified from Supplementary Table 1 to Table S1.

Second response to reviewer 1 in blue below, responding to second comments in red following initial response in green.

1. It would be nice if the authors stated the overall importance of this finding. Do the structural details they find have implications on nanoparticle behaviour? Do they change the critical stage of endosomal escape in transfection, for which efficacy are estimated to be as low as 2%. Can this be clarified?

This paper has a novel method development focus. We would not want to overstretch the interpretation of our data to the complex environment of endosomal release and its poor efficacy based upon one LNP composition. We have clarified this at the end of the discussion: The new analytical insights this novel method can achieve can be paired with endosomal mimics to examine the influence on LNP components in endosomal escape, offering a powerful framework for to improve therapeutic formulations. 34

The reviewer understands the focus of the paper on method development, and it is good to add this statement at the end of the discussion. Please refine the sentence as it is not clearly written (highlighted in the manuscript).

Thank you for this observation, we have improved the language.

2. The Summary is poorly written with grammatical and punctuation errors and seems almost rushed. It is also not a very clear statement of what is in this manuscript.

This has been modified, but retains the aim of being a high level description of the method and its potential future application to complement the details that has been presented previously.

It is good that the authors have made modifications. There are still some unclear phrases and typos in the summary (highlighted in the manuscript). Please revise.

Thank you for this observation, we have improved the language.

3. Figure S3, can the authors indicate why uracil is not observed in the SIMS spectra?

As noted on page 6 this is "most likely due to significantly lower secondary ion formation probability".

It is more likely that the uracil is under the detection limit possibly due to its low concentration in the RNAs compared to the other nucleotide bases (abundance of different nucleotides in RNA can be different according to the type of the RNA and its function), not because of its ion formation probability.

Figure S3 Cryo-OrbiSIMS profiles of Formulation 1 displaying profiles from RNA fragments from adenine, guanine and cytosine along with the PEG ions. Uracil was not seen in the spectra. The intensities have been normalised to total ion count and all RNA fragment ions are presented on secondary intensity scale for clarity.

This is an interesting suggestion regarding the abundance in the RNA being the dominant contributor to the signal intensity. First, we must note that the RNA is produced by vitro transcription which will use only 5-methoxy UTP present in the reaction, something we had not previously stated. The methods has been updated to clarify that thus:-

"The RNA cargo was comprised of a 1:1 mol:mol of the following constructs: CleanCap® Firefly Luciferase mRNA (5-methoxyuridine) and CleanCap® Erythropoietin mRNA (5-methoxyuridine) formulated in a 50mM citrate buffer at pH 3.0. "

Consequently, we checked our spectra for the 5-methoxyuridine representative ion, which was not found. This is now clarified in the manuscript. Its structure is listed below along with the structures and characteristic ions that are detected for adenine, guanine and cytosine (now added to Fig S3).

We checked the composition of the coding portion of the mRNA formulation as the best available estimate of the oligonucleotide composition, and found that 5-methoxyuridine is the least abundant RNA nucleotide, at 14 mol% (calculation below). However, adenine at 18 mol% is only slightly more abundant yet has a clear signal which is plotted versus depth in Fig 3S, which is far greater than the noise. Indeed, we can see that adenine is the most intense signal in Fig S3, and the relative amounts at the nucleotide intensity plateau in the depth profile in Fig S3, do not follow the relative abundance of the nucleotide C>G>A>U (below). Rather it is A>G>C with U=0. This supports our initial proposal that different ionisation probabilities are the most likely reason that 5-methoxyuridine is not detected. We have therefore retained this hypothesis on page 7, and expanded it to include the likely role of the primary amine structure in the higher secondary ion yield of the other nucleotides. We hope with this new information, the reviewer will agree that this is the most likely cause of this observation.

"The adenine profile is presented to reveal the location of the mRNA molecules in Figure 1b. Guanine and cytosine are follow similar profiles as adenine (Figure S3). A 5-methoxyuracil ion, representative of 5-methoxyuridine, was not detected which is most likely due to significantly lower secondary ion formation probability. The 5-methoxyuracil fragment is the only nucleotide without a primary amine group, which is the likely charge centre for the adenine, guanine and cytosine secondary ions."

Supporting calculations

1:1 mol:mol mixture of:-

fLuc: U=0.14 A=0.20 G=0.31 C=0.34

(sequence sourced from https://www.trilinkbiotech.com/media/maravai/productattachments/product_insert/fluc_orf_catno_I-8102_I-7202_I-7602_.txt)

EPO: U=0.13 A=0.16 G=0.35 C=0.36

(sequence sourced from https://www.trilinkbiotech.com/media/maravai/productattachments/product_insert/epo_orf_catno_I-8109_I-7209_.txt).

Thus, the average nucleotide composition is U=0.14 A=0.18 G=0.33 C=0.35

Response to Chemical Communications reviewers' comments

We thank the reviewers for the time and close attention they have given to the manuscript with their expertise. It will certainly improve it. We have made every effort within our capability to address the points made by the reviewers below (in green text following their comments), and with tracked modifications to the submitted manuscript.

Reviewers' comments:

Reviewer #1

Thank you for answering the questions. Please find the reviewer 1's responses below in red.

It would be nice if the authors stated the overall importance of this finding. Do the structural details they find have implications on nanoparticle behaviour? Do they change the critical stage of endosomal escape in transfection, for which efficiencies are estimated to be as low as 2%. Can this be clarified?

This paper has a novel method development focus. We would not want to overstretch the interpretation of our data to the complex environment of endosomal release and its poor efficacy based upon one LNP composition. We have clarified this at the end of the discussion:

The new analytical insights this novel method can achieve can be paired with endosomal mimics to examine the influence on LNP components in endosomal escape, offering a powerful framework for to improve therapeutic formulations.³⁴

The reviewer understands the focus of the paper on method development, and it is good to add this statement at the end of the discussion. Please refine the sentence as it is not clearly written (highlighted in the manuscript).

The graphical abstract is complicated and still does not get across the critical message of the manuscript.

When cryo TEM is used this should be indicated. The authors write in the Results, "Samples were transferred to the OrbiSIMS instrument under liquid nitrogen after TEM imaging for molecular depth profiling under cryogenic conditions (-170°C)." making it seem they used cryo TEM, but it is not clearly stated.

Thank you-corrected.

It does not seem that DLS and PDI are defined.

Thank you-corrected.

The authors write that the guanine and cytosine profile across the nanoparticle are similar. Do they mean to adenine and is it possible to perhaps include these data to be more convincing?

Yes, similar to adenine which has been clarified. These data are all in Figure S3, the indication of which has been moved forward in the sentence to make it clearer.

Missing an “and” in the sentence, “While these two systems are very different cannot to be compared directly,”

Thank you-corrected.

Figure 2b has five ions listed, but only four colors and lines in the plot.

Thank you-corrected.

Furthermore, the data are only really convincing at depth to about 5 nm. The authors acknowledge this but it might be made a little more clear as the deeper signals should be the adenine showing the RNA core and this is discussed a great deal in the manuscript.

The authors feel that it is important to clarify that the thickness below which the matrix effect is minimised is 5 nm. We discuss this in the last paragraph on page 11 regarding the limitations of the depth estimates in this early stage of such a novel approach. The depth resolution will degrade with depth, but for this sample format this remains to be quantified. However, we do have confidence that the adenine and other RNA signals are reliably generated from the LNP core.

The paragraph on AI seems out of place and unconnected to the rest of the manuscript.

We would argue that the paragraph is very relevant to the role this analysis technique will likely assume given the role of AI in LNP discovery as well as other biomaterials. We placed it directly after the section describing recent high throughput advances, but have reduced it and tried to signpost it better in this resubmitted manuscript.

It is good that the authors modify the paragraph and relate the AI to high throughput LNP discovery now to make it sound more connected to the work.

The Discussion can be shortened. Some of this is rehash of the Introduction and some seems unrelated and should either be put in context to the work presented here or omitted.

We have considered all specific points by the reviewers and made changes in the discussion noted throughout. Given this is the first sight of this data type we think a significant discussion of the data here and how this method needs to develop is warranted. If the reviewer has some more specific requests we can address these, as the AI one above where are reduction was made.

The Summary is poorly written with grammatical and punctuation errors and seems almost rushed. It is also not a very clear statement of what is in this manuscript.

This has been modified, but retains the aim of being a high level description of the method and its potential future application to complement the details that has been presented previously.

It is good that the authors have made modifications. There are still some unclear phrases and typos in the summary (highlighted in the manuscript). Please revise.

Is Figure S1a really just a straight line?

Figure S1a is a table. We have modified it to clarify this.

Figure S3, can the authors indicate why uracil is not observed in the SIMS spectra?

As noted on page 6 this is “most likely due to significantly lower secondary ion formation probability”.

The reviewer is not convinced by this statement. Looking at the molecular structures of all four nucleotide bases, the structures of adenine and guanine are similar whereas those of uracil and cytosine are similar. Here, the authors could detect three of the bases including cytosine. It is more likely that the uracil is under the detection limit possibly due to its low concentration in the RNAs compared to the other nucleotide bases (abundance of different nucleotides in RNA can be different according to the type of the RNA and its function), not because of its ion formation probability.

Reviewer #2 (Remarks to the Author):

The manuscript of Kotowska et al. presents a new methodology for the spatially resolved characterisation of lipid nanoparticles. Lipid nanoparticles are very important i.e. as carrier for drugs. Usually the local structure of the nanoparticles is obtained by neutron scattering. Here the authors use high resolution mass spectrometry (OrbiSIMS) for the elucidation of the composition from core to shell. This approach is highly innovative and as the presented results provide important and precise information about the 3D composition. It is worth to be reported in "Nature Communications in Chemistry". However I have three major points that should be clarified / improved before publication.

1) Within the manuscript no information, obtained by established methods, about the structure of the nanoparticles is given. For such work, I would expect that an established method like SANS is used to initially characterize and later discuss the reliability and added value of the new method. The shown TEM data give no hint to the lipid structure.

We have specifically chosen a well-researched LNP system, for which the DMG-PEG structure is universally agreed upon in the literature. In no publications is the external orientation of PEG in the DMG-PEG disputed-indeed it is required to produce and stabilise monodisperse LNPs. It is this orientation that we use to validate our observation with the novel Cryo-OrbiSIMS technique. The PEG thickness was quantified in Arteta et al 2018 using SANS, although this is not required to validate our main finding, i.e. we can detect molecular orientation using Cryo-OrbiSIMS depth profiling. Undertaking small angle neutron scattering (SANS) analysis is a very major undertaking, not least the need to apply for and schedule beam time, and we do not regard that replicating the well regarded Arteta paper in this paper would be beneficial to our main claim.

Our LNP production method was validated morphologically using cryo-TEM and DLS. The observation that the particles are physically as to be expected gives us confidence that the LNP structure is as reported in the literature. This is consistent with our OrbiSIMS findings, which for the DMG-PEG orientation are very clear.

2) It is not clear whether the analysis was repeated on several samples. Please state clearly in the experimental section how many repetitions were carried out.

The number of replicates was noted in the results section, and has now been duplicated in the methods section to satisfy this request.

3) The nanoparticles have a spherical structure. Therefore SIMS depth profiling data origins from different height levels and in the depth profile information from different areas is mixed. Please discuss / comment on the effect of the geometry with respect to the analysis.

The effect of particle geometry on SIMS is expanded upon briefly on page 11 in response to this request. Briefly, the height of the particle is not of importance, rather that the particles' strata are sampled sequentially as etching proceeds. The exception of shadowing and preferential sputter yields discussed in the Yang et al. paper is now cited along with the inclusion of an explanation of this concept in the methods section:

Etching removes the particle strata which are sampled sequentially as primary ion bombardment proceeds, with the exception of shadowing and preferential sputter yields discussed in the Yang et al.³⁹

I have also a few minor comments:

1) The arrows with the text inside are not so nice for a graphical abstract. Maybe the SIMS process can be better illustrated by a illustration with a beam and "flying ions".

We agree and have modified this.

2) L. 126: The abbreviation "PDI" is not introduced.

Thank you-corrected.

3) L.188/189: Something is missing in the first part of this sentence. What cannot be determined?

We presume the reviewer means L243 where the word and was omitted: '...are very different and cannot be compared...' which has been corrected with the addition of the word "and".

4) Fig. 1b: It is not clear for the reader which signal intensities are belonging to the second intensity axis.

This is noted in the caption-and is now also clarified in the graph itself using arrows for those on the secondary axis.

5) Fig. 2: It is really hard to corelate which masses belongs to which intensity scale. Maybe this information can be added to the graphs with arrows?

This is noted in the caption-and is now also clarified in the graph itself for those on the secondary axis.

6) It is quite unusual that a pure Ar3000+ beam is produced with such a high current. Ususally, in the IONTOF GCIB sources a beam with a cluster distribution is used. Please comment on this and give the cluster distribution in SI.

This is correct, the GCIB beam is not pure Ar 3000+ but is normally abbreviated as such. The cluster distribution is now noted in Figure S5.

7) L. 381. The given area should have the dimension of an area (μm^2 instead of μm). Please check also the following text.

This has been corrected to $400 \mu\text{m} \times 400 \mu\text{m}$ rather than μm^2 to avoid the potential confusion between $400 \mu\text{m} \times 400 \mu\text{m}$, $400 \mu\text{m}^2$ and $160,000 \mu\text{m}^2$

8) L. 385: I doubt that an analysis is possible with such a high Ar pressure in the main chamber. Please check the dimension.

Argon gas flooding was in operation in order to aid charge compensation, with pressure in the main chamber maintained at 9.0×10^{-7} mbar.

Reviewer #3 (Remarks to the Author):

The results are convincing and the conclusions are well founded. However, my opinion is that the authors need to consider the following points in order to improve the quality and impact of the paper.

My major concern relates to the hydrated state of the LNPs during the OrbiSIMS analysis.

The authors state that the method relies on the retention of the LNPs in the hydrated state. However, I would like to see evidence that this is actually the case during the OrbiSIMS analysis. The TEM image acquired prior to SIMS analysis indeed verifies structural integrity, but can the authors verify that the sample is hydrated also after transfer into the OrbiSIMS instrument? If the sample is hydrated (i.e., embedded in ice), I would expect to see data for ions associated with water, such as $(\text{H}_2\text{O})_n\text{H}^+$ and/or $(\text{H}_2\text{O})_n\text{Na}^+$ cluster ions. Can the authors show depth profiles for such water-associated ions?

1. The vision of “embedding” of the particles is understandable given the frozen hydrated nature of the sample, but is in fact a misconception. The below situation is instead expected due to optimisation of the blotting time, the surface tension of the water and the geometry and relative dimension of the particles and grid components:-

Embedding (ice overlayer):

Supported frozen particles incorporating water:

Figure S1c) Frozen nanoparticle schematic for the Cryo-OrbiSIMS analysis..

This arrangement is consistent with there being LNPs represented in the first spectra acquired before etching commences. If this was pure ice, they would not be present and would be observed later in the depth profile. We propose including this in the SI, with the description:

“Lipid secondary ions are seen in the first spectrum acquired from each sample, before etching commenced, consistent with the view of frozen particles supported by the 10 nm thick amorphous carbon support film, rather than particles embedded in ice (Figure S1).”

The absolute intensities of the lipid nanoparticle components at the first scan of the analysis are:

Peak assignment	Component name	Intensity [a.u.]
Summed ions	FA 14:0 summed	0
Summed ions	PEG summed	7436.56
$C_{27}H_{45}^+$	cholesterol	10023.16
$C_{27}H_{46}ONa^+$	cholesterol	37215.19
$C_{43}H_{80}NO_2^+$	DLin-MC3-DMA	984544.03
$C_{44}H_{88}NO_3P^+$	DSPC	21577.38
$C_5H_6N_5^+$	RNA adenine	0

2. The Orbitrap detector with SIMS does not include water ions as water clusters in the same way as in ToF SIMS. This is likely due to the delicate water complexes falling apart whilst awaiting injection into the Orbitrap, or during the argon cell used to reduce the ion energy to allow their acceptance by the OrbiTrap analyser. The molecular water ion, OH_2^+ appears at a mass below the minimum for inclusion into the Orbitrap spectrum. However, we do have ToF SIMS analysis

Commented [MA1]: @Anna Kotowska (staff) to provide non normalised first and second spectra

Commented [DS(2)]: I might dispute this concept. First analysis scan is presumably above the PIDD.

from additional experiments that we did not previously include, giving us confidence that the samples are frozen. This has been inserted into the SI as below:

Figure S1 ...d) LMIG ToF surface analysis spectra of the LNP formulation acquired in cryogenic conditions. The repetitive pattern represents water cluster peaks, the first 3 water cluster peaks are assigned in the spectrum, with the other indicated by asterisk.

Additionally, the use of Na₂Cl⁺ to represent Na in the buffer is not very convincing, if we assume that the LNPs are embedded in ice. Instead, ions of the type (H₂O)_nNa⁺ would be a better choice. Can the authors provide depth profiles for ions that more realistically can represent buffer-associated Na?

The buffer is the only component that includes significant Na₂Cl⁺ intensity as can be seen in Figure S2. There are no water-Na⁺ ions in the ToF spectrum now presented in Figure S1d. The Na₂Cl⁺ ion is not normally considered to be controversial as an indicator of sodium chloride for those used to dealing with Orbitrap data, but we have inserted a clarification on the point of water clusters.

“Water clusters from the aqueous suspension are not seen in the OrbiSIMS spectra, a phenomena common across all frozen hydrated work, rationalised as the instability of the water clusters prior to Orbitrap analysis.”

Finally, to explain why the adenine depth profile shows a gradual increase and not a sharp step at the interior of the LNP, the authors argue that an inhomogenous thickness of the PEG layer on the LNP surface would lead to such a gradual increase. This is correct if PEG is the only component on the LNP surface, i.e., if the LNPs are not embedded in ice. If embedded in ice, the sputter time needed to reach the mRNA would not be greatly affected by the thickness of the PEG layer. Other possible explanations for the gradual increase could be the spherical topography of the LNPs and/or inhomogeneous spatial distributions between different LNPs.

So, the question is how the authors envision the hydrated state. Are the LNPs embedded in a thin film of ice (as typically envisioned for the cryo-TEM preparation used in this work) or has the bulk ice disappeared, with the structural integrity of the LNPs stabilized by the low temperature and/or individual water molecules directly bound to the lipid structures. Without evidence for bulk ice in the sample, the latter possibility seems more likely to me. A more detailed discussion about the “hydrated state” would definitely be beneficial and clarifying for how to interpret the data. Additionally, transfer back to the TEM and cryo-TEM analysis after cryo-OrbiSIMS would provide conclusive evidence for the structural integrity of the LNPs during the cryo-OrbiSIMS analysis.

As noted above, LNP components appear in the first spectra. Hopefully this is now clear to the reviewer with the addition of the ToF SIMS and the explanation of the geometry in Figure S1c.

Detailed comments:

The manuscript contains numerous obvious errors and some unclear formulations that need attention by the authors, some of which are listed below. In addition, I recommend the authors to make a detailed review of the entire manuscript (text and figures) to find and correct possible additional errors.

We thank the reviewer for pointing out these typos, they have been attended to, and the manuscripts has been additionally proofread.

Page 3, "The also enable...": A word seems to be missing

Page 4 "...to directly analysis the nanoparticle..." Should be "...analyse..."?

Page 5 "(Figure S1)", two places Should be "(Figure S2)"?

Page 5 The $CxH_{2x}O_{x/2+1}$ etc formulas need to be checked, do not agree with the mentioned examples, also in Figure S2

This has been corrected and the table s1 expanded to clarify the structural assignment further.

Page 5 "A complete list is provided in Table S1" The list is not complete, contains only the PEG fragments. Change text to "A list of the PEG fragments..." or add the FA14:0 ions to the list.

The caption stipulated that the DMG FA fragments were not included. We have not added the FA(14:0) fragments to the list.

Page 6 "...levels off to approximate to the core of the particles..." Unclear

Clarified thus: "interpret the depth at which the adenine signal levels off to represent the depth at which the core of the particles has been reached."

Page 6 "...in the first analysis of the profile..." Unclear, change "analysis" to "data point", or similar

Clarified to: "There is near zero intensity of the DMG molecules in the first spectrum acquired, represented in the first data point of the profile"

Page 6 "...results in a the RNA intensity..." Remove "a"?

Done.

Page 7 "...which, although extending this beyond this initial proof-of-concept to more LNP formulations, the PEG thickness variability will likely be able to be correlated anticipated LNP structure, e.g. by varying the PEGylated lipid amount." I don't understand the meaning of this, reformulate.

To clarify we have expanded to "It is not possible using this approach to determined which of these alternative scenarios is occurring, beyond this initial proof-of-concept, the PEG thickness variability will likely be able to be correlated and anticipated from varying the LNP formulation, e.g. by modifying the PEGylated lipid amount."

Page 7, Figure 1 caption References 31, 33 seems incorrect to me, check.

Thank you-these are now re-joined to the reference manager and display the correct citation.

Page 8 Last sentence starting “While these two systems...” contains at least two errors and is otherwise also very difficult to decipher. Correct and reformulate.

Thank you. Sentence split into 2 to make clearer: “While these two systems are very different and cannot to be compared directly, the observation of the different cholesterol fragments suggests molecular orientation in a model pure crystalline system. This is consistent with our observation of molecular orientation of the larger PMG-PEG2k using GCIB etching in this LNP system, suggesting that the orientation of small molecules such as cholesterol may be detectable in some systems.”

Page 9, Figure 2b The labels include four trace colors but five assignments, one assignment must be removed

This has been corrected.

Page 12 “...is critical engineering their performance.” A word is missing?

Thank you: “is critical for engineering their performance”

Page 13 RiboGreen RNA assay kit Add manufacturer/reference

This clarification has been inserted: “Invitrogen™ Quant-it™ RiboGreen Reagent and RNA Assay Kit (Thermo Fisher)”

Page 15 The entire paragraph starting “Each depth profile was acquired...” does not make sense to me, do not understand. Reformulate.

Reformulated to be clearer:

Each depth profile was acquired from the full thickness of the sample. An initial nanoparticle components region was first seen, followed by a complex mixture of inorganic ions (not shown) interpreted to be from the surface of the carbon support substrate were found, appearing in the profile below the particles before the substrate was reached. We do not interpret the relative positions of these ions, instead focussing on the organics which represent the lipids.

Page 18, Figure S2 “C54H89NPO8+” Should be “C44H89NPO8+”?

Thank you – corrected.

Page 22 “Table 1” Should be “Table S1”?

We have modified from Supplementary Table 1 to Table S1.

Mass spectrometry method development collection

Title: Molecular orientation and stratification revealed in RNA-lipid nanoparticles using Cryogenic Orbitrap Secondary Ion Mass Spectrometry (Cryo-OrbiSIMS) depth profiling

Anna M Kotowska¹, Michael Fay², Julie A Watts², Ian Gilmore⁶, David J. Scurr¹, Alaina Howe⁵, Vladimir Capka⁵, Corey Perez⁵, Devin Doud⁵, Siddharth Patel⁵, Mark Umbarger⁵, Robert Langer^{3,4}, Morgan R Alexander¹

1. School of Pharmacy, University of Nottingham, Nottingham, UK.
2. Nanoscale and Microscale Research Centre, University of Nottingham, Nottingham NG7 2QL, UK.
3. Department of Chemical Engineering, Massachusetts Institute of Technology, Cambridge, MA, USA.
4. Koch Institute for Integrative Cancer Research, Massachusetts Institute of Technology, Cambridge, MA, USA.
5. Sail Biomedicines, Cambridge, MA, USA.
6. National Physical Laboratory, Hampton Rd, Teddington TW11 0LW, UK.

Abstract

Lipid nanoparticle RNA (LNP-RNA) formulations are used for the delivery of vaccines and other therapies. RNA molecules are encapsulated within their interior through electrostatic interactions with positively charged lipids. The identity of the lipids that present at their surface play a role in how they interact with and are perceived by the body and their resultant potency. Here, we use a model formulation to develop cryogenic sample preparation for molecular depth profiling Orbitrap secondary ion mass spectrometry (Cryo-OrbiSIMS) preceded by morphological characterisation using cryogenic transmission electron microscopy (Cryo-TEM). It is found that the depth distribution of individual lipid components is revealed relative to the surface and the RNA cargo defining the core. A preferential lipid orientation can be determined for the 1,2-Dimyristoyl-glycero-3-methoxy-polyethylene glycol 2000 (DMG-PEG2k) molecule, by comparing the profiles of PEG to DMG fragments. PEG fragments are found immediately during analysis of the LNP surface, while the DMG fragments are deeper, coincident with RNA ions located in the core, in agreement with established models of LNPs. This laboratory-based de novo analysis technique requires no labelling, providing advantages over large facility neutron scattering characterisation.

Deleted: We

Deleted: with

Deleted: at

Graphical abstract

Introduction

Billions of doses of mRNA-loaded lipid nanoparticle (LNP) vaccines represent a spectacular recent example of the utility of therapeutic nanomaterials.¹ The molecules presented on the surface of such LNPs influence how the material interacts with the environment inside the body,² including which biomolecules it adsorbs to form the biomolecular corona.³ This *biointerface* is critical in determining nanoparticle trafficking, potency and immunological reaction, yet uncertainties surrounding LNP structural organisation, such as the location of ionisable lipids either in the core^{4,5} or in the exterior shell, remain.^{6,7,8,9} Such structural details likely have implications on nanoparticle behaviour, including the critical stage of endosomal escape in transfection, for which efficiencies are estimated to be as low as 2%.^{10,11}

Synthetic LNPs are commonly composed of four components; an ionizable lipid, a helper lipid, cholesterol and a poly(ethylene glycol)ylated (PEG) lipid. Ionisable lipids interact electrostatically with the negatively charged mRNA facilitating its encapsulation within the LNP. These also enable endosomal escape by appropriate pKa choice. Helper lipids, such as the phospholipids distearoylphosphatidylcholine (DSPC) and dioleoylphosphatidylethanolamine (DOPE), contribute to the structural integrity and stability of LNPs. Cholesterol is incorporated into LNPs to modulate their fluidity and rigidity, enhancing particle stability and membrane fusion capabilities and also helps maintain the structural integrity of LNPs, allowing for delivery and release of mRNA into target cells. The inclusion of PEG-lipids on the LNP surface contributes to both formulation stability,¹¹ and the 'stealth' properties of the nanoparticles,^{5,12} with longer fatty acids allowing them to circulate in the bloodstream longer and increasing the chances of reaching their target cells whilst shorter fatty acids are shed faster enabling the lipid surface to be exposed and absorb the serum proteins.¹³

The study of the structural, compositional, and functional aspects of LNPs for drug delivery has employed various analytical and imaging techniques. Cryo-EM allows for the observation of the nanoparticle morphology and internal structure, providing insights into the lipid organization, but without information on molecular identity.^{14,15} Small-angle neutron scattering (SANS) combined with deuteration of components at large scale facilities has been used to quantify the lipid distribution at the surface of LNPs using isotopic contrast variation.⁵ Small-angle X-ray scattering (SAXS) using synchrotron sources, has been used to probe bulk phase structures.¹⁶ Dynamic light scattering (DLS) is used to rapidly measure the size distribution and polydispersity index (PDI) of LNPs, offering insights into their

Moved down [1]: We find this laboratory-based direct analysis method can identify the molecular structure of the LNP surface, revealing the relative depths of lipid and RNA components. Furthermore, the orientation of certain molecules can even be discerned where fragments resulting from different portions of the molecule are detected.

Deleted: To provide a method that can address these uncertainties, we here show that molecular depth profiling of LNPs is possible using cryogenic OrbiTrap™ secondary ion mass spectrometry (Cryo-OrbiSIMS).

Deleted: This information relies on retention of the native hydrated structure in ice within the vacuum of the Cryo-OrbiSIMS instrument, verified by analysis using transmission electron microscope (TEM) on the same sample....

Deleted: The

Deleted: Study

homogeneity and stability. NMR spectroscopy provides information on the molecular structure, dynamics, and interactions within LNPs.¹⁷ It can be used to study the composition of the lipid core, the conformation of PEG chains, and the interaction between PEG and other components of LNPs, however the structural analysis of LNPs is limited by NMR sensitivity.¹⁸

Deleted: its

To provide a method that can address these uncertainties, here we show that molecular depth profiling of LNPs is possible using cryogenic OrbiTrap secondary ion mass spectrometry (Cryo-OrbiSIMS). We find this laboratory-based direct analysis method can identify the molecular structure of the LNP surface, revealing the relative depths of lipid and RNA components. Furthermore, the orientation of certain molecules can even be discerned where fragments resulting from different portions of the molecule are detected. This information relies on retention of the native hydrated structure within the vacuum of the Cryo-OrbiSIMS instrument, verified by analysis using cryo-transmission electron microscopy (Cryo-TEM) on the same sample.

Moved (insertion) [1]

Formatted: Font colour: Black

Vacuum-based surface chemical analysis techniques such as X-ray photoelectron spectroscopy (XPS) or time of flight secondary ion mass spectrometry (ToF-SIMS) have the capability to analysis the nanoparticle surface to describe it in elemental or molecular terms respectively. A recent study reported the possibility of evaluating the surface chemistry of lipid nanoparticles using XPS, necessitating cryogenic conditions, including the detection of PEG on the surface.¹⁹ Extracellular lipid vesicles have also be imaged using ToF-SIMS, however the chemical information obtained from the spectra is not sufficient to identify the different component molecules.²⁰ While depth profiling has previously been employed to characterize the internal structure of giant (10-30 μm) liposomes, it's application toward depth-profiling of lipid particle molecular identification at the nanoscale has not been reported.²¹ The development of OrbiSIMS addresses the limitations of ToF-SIMS in analysing high-mass biomolecules with high mass resolving power, whilst utilising gas clusters as an analysis beam provides improved detection of high molecule weight species and preserves molecular information when depth profiling.²²

Deleted: directly

Deleted: can

Deleted: ToF-SIMS

Deleted: ToF-SIMS

Results

A cryogenic preparation protocol ~~was first~~ developed to examine LNPs in vitreous ice sequentially on different areas of the same sample by ~~Cryo-~~TEM and then Cryo-OrbiSIMS. To test the capability of ~~Cryo-OrbiSIMS depth~~ profiling for the characterisation of the surface and subsurface chemistry of the LNPs, we analysed a model formulation comprising mRNA and 4 lipids; DMG-PEG2k, cholesterol, Dlin-MC3-DMA as the ionisable lipid and DSPC as the helper lipid (Table 1) formed using microfluids. The nanoparticle suspension in phosphate buffered saline (PBS) was deposited onto a plasma etched 10 nm thick amorphous carbon film with regularly spaced 2.4 µm diameter holes supported by copper TEM grids. These were blotted to modulate thickness before plunging into liquid ethane, after which they were transferred under liquid nitrogen for ~~Cryo-~~TEM imaging (-172°C). The particles were found by ~~Cryo-~~TEM to be relatively monodisperse, with their diameters distributed around a mean of 60-80 nm (Figure 1a) while with DLS they were estimated to be slightly larger with a mean diameter of 91 nm and a PDI of 0.03 (Figure S1).

Samples were transferred to the OrbiSIMS instrument under liquid nitrogen after TEM imaging for molecular depth profiling under cryogenic conditions (-170°C). An Ar₃₀₀₀⁺ gas cluster source was used both as a primary ion analysis and depth profiling beam. ~~Data was acquired from~~ an ensemble of particles simultaneously over a 400 µm x 400 µm area. ~~Since~~ the lateral resolution of the analysis beam is not sufficient to resolve individual LNPs, ~~this approach aimed to maximise the secondary ion count from the LNPs available from such a limited amount of material. Secondary ions diagnostic of lipids were observed in the first spectrum acquired from each sample, consistent with the view of frozen particles supported by the 10 nm thick amorphous carbon support film, rather than particles embedded in ice (Figure S1). Water clusters from the aqueous suspension are not seen in the OrbiSIMS spectra, a phenomenon common across all frozen hydrated analysis, rationalised as the instability of the water clusters prior to Orbitrap analysis. A ToF-SIMS analysis from the surface of the sample before etching (Figure S1d) shows water clusters, confirming the hydration of the LNPs, consistent with the model of frozen hydrated LNPs sitting on the carbon support, rather than the embedded alternative shown in Figure S1c.~~ Secondary ion assignments were guided by the individual component reference spectra (Figure S2), although in the case of this LNP formulation, all observed fragments were unique to one component ~~and definitively linked to the lipid molecular structures.~~

Cryo-OrbiSIMS molecular profiles: DMG-PEG

Deleted: is

Deleted: cryogenic OrbiSIMS

Deleted: and

Deleted: cryo

Deleted: These measurements were undertaken on

Deleted: since

Deleted: .

Deleted: S1

Deleted: ,

Since the exterior orientation of the PEG group is well established in the literature for LNPs containing DMG-PEG2k, we first examine the secondary ions from this molecule. Both DMG fragments and sodiated PEG2k fragment ions defined in the standards (Figure S2) were detected in the SIMS spectra and are plotted as a function of etch time and estimated depth in Figure 1b (2 further replicates are presented in Figure S4). The PEG ion fragments in the LNPs are all sodiated and of the form $\text{Na}[\text{O}-\text{CH}_2-\text{CH}_2]_x^+$, $\text{Na}[\text{O}-\text{CH}_2-\text{CH}_2]_x-\text{O}-\text{CH}_2^+$ or $\text{NaCH}_2[\text{O}-\text{CH}_2-\text{CH}_2]_x-\text{O}-\text{CH}_2-\text{CH}_2^+$. The DMG anchor is detected as the whole molecular fragment: $\text{C}_{31}\text{H}_{59}\text{O}_4^+$, and by ions representing its FA14:0 moiety: $\text{C}_{14}\text{H}_{27}\text{O}_2^+$, $\text{C}_{14}\text{H}_{27}\text{O}_2\text{Na}^+$. A complete list is provided in Table S1.

Deleted: S1

Deleted: versus

Deleted: $\text{C}_x\text{H}_{2x}\text{O}_{x/2+1}\text{Na}^+$ (e.g. $\text{C}_{23}\text{H}_{46}\text{O}_{12}\text{Na}^+$, $\text{C}_{24}\text{H}_{48}\text{O}_{13}\text{Na}^+$)...

Formatted: Font colour: Black

Deleted: $\text{C}_x\text{H}_{2x}\text{O}_{x/2+2}\text{Na}^+$ (e.g. $\text{C}_{25}\text{H}_{50}\text{O}_{13}\text{Na}^+$).

Formatted: Subscript

There was a high intensity of PEG signal at the beginning of the analysis which decreased with etching, consistent with the expectation that the PEG portion of the molecule orients outwards for the LNPs. Notably the PEG profile, which is constructed from sodiated PEG ions, does not correlate with the chloride profile derived from the Na_2Cl^+ ion (Figure 1b). This suggests that the sodiated-PEG adduct ions are reflective of an ionic association in the hydrated sample which is an established phenomenon for PEG,²³ and that the intensity of these ions is reflective of the amount of PEG in the sample, rather than the Na_2Cl^+ distribution. Cationization of PEG through doping prior to SIMS analysis has previously been used to increase the secondary ion yield of oligomers.²⁴ However, in our experiments the sodium from the PBS particle suspension has enhanced the yield of the PEG from the frozen sample and represents the actual hydrated sample chemistry rather than an artefact of preparation. Notably the PEG in the DMG-PEG2k component spectra in Figure S2 is also dominated by sodiated secondary ion adducts.

Deleted: sodium

Deleted: artifact

There is near zero intensity of the DMG molecules in the first spectrum acquired, represented in the first data point of the profile indicating that the fatty acid hydrocarbon chains are not present at the outermost surface, fitting the standard description that they are oriented inwards. The PEG and FA14:0 profiles cross at approximately the middle of each ion intensity range at 4-7 nm depth. This measurement can provide an estimate of the thickness of the PEG2k shell. The value is similar to the SANS estimate of a 4 nm thick surface layer of PEG2k assumed to be in a mushroom conformation for similar PEG2k containing particles.⁵ A direct comparative analysis of SANS on the same particles would be needed to make a more accurate comparison, but for the purpose of this method development study, the similarity of the value acquired from Cryo-OrbiSIMS depth profiling to that from SANS value reported in previous studies provides a degree of cautious confidence in the secondary ion profile estimated depth values.⁵

Deleted: analysis

Deleted: picture

Deleted: -

Deleted: proof-of-concept

Deleted: SIMS

Deleted: the literature

The adenine profile is presented to reveal the location of the mRNA molecules in Figure 1b. Guanine and cytosine are also seen to follow similar profiles as adenine (Figure S3), although uracil was not detected, most likely due to significantly lower secondary ion formation probability. The adenine ion intensity had near zero intensity at the surface, which increased at a constant rate until ca 8 nm depth when it plateaued. We interpret the depth at which the adenine signal plateaus to represent the depth at which the core of the particles has been reached. The observation of adenine intensity before 8 nm is reached, rather than a sharp step up from zero to the value in the core, is interpreted as reflecting a range of PEG overlayer coverage, with the thinner coverage being ablated by sputtering more quickly, whilst thicker areas take longer to sputter through to reach the adenine. This results in the RNA intensity from the core increasing gradually over a range of depths. The DMG ions were equally seen to be revealed gradually, rather than as a step change, consistent with the PEG profile. Since this is an ensemble measurement, this range in PEG coverage may either be within individual particles, or it may represent different particles with different PEG thickness. It is not possible using this approach to determine which of these scenarios is occurring. The molecular orientation findings are summarised in a simplified schematic of the molecular structure for DMG-PEG2k in Figure 1c to illustrate the interpretation of the data, but this does not depict the PEG thickness variation inferred from the profile breadth.

Deleted: ,

Deleted: (Figure S3).

Deleted: levelled off

Deleted: levels off

Deleted: approximate to

Deleted: particle.

Deleted: taking

Deleted: .

Deleted: a

Deleted: determined which, although extending this beyond this initial proof-

Deleted: -concept to more LNP formulations, the PEG thickness variability will likely be able to be correlated anticipated LNP structure, e.g. by varying the PEGylated lipid amount...

Deleted: include

Figure 1 Comparison of the morphology (Cryo-TEM) and chemical composition (Cryo-OrbiSIMS). a) TEM of the LNP formulation with histograms of particle size. b) Cryo-OrbiSIMS depth profiles with cross sectional schematics of LNP component distribution. The intensity has been normalised to total ion count and the summed FA14:0 ions are presented on secondary intensity scale (indicated by arrow) for the depth profile clarity. The secondary x axis presents depth estimated from comparison with organic standards. Further replicates are included in Figure S4. c) Simplified schematic of DMG-PEG2k to show the relative molecular orientations.

Deleted: PEG

Deleted: for

Deleted: 31,33

Formatted: Font colour: Black

Deleted: simplified

Ionisable lipid, helper lipid and Cholesterol

In Figure 2, we present profiles for the DSPC, Dlin-MC3-DMA and Cholesterol lipid components. All DSPC fragments were seen to monotonically increase from the analysis of

Deleted: ,

the outermost surface as profiling was undertaken, although not from zero intensity (Figure 2a). Adenine representing the RNA core also increased but begins at zero, suggesting that DSPC is present in the shell as well as the core. There is no clear orientation indicated by comparing the headgroup and fatty acid fragments which have similar profiles.

Deleted: profile shapes

The profile of the ionisable lipid Dlin-MC3-DMA (Figure 2b) indicated that it was strongly represented at depths less than 3 nm where PEG is also present (< 8 nm), thereafter decreasing in intensity and recovering at depths greater than 6 nm. This may represent a propensity for surface segregation, or an association of Dlin-MC3-DMA with the PEG at the exterior, similar to the sodium and chloride ions from the PBS.

Deleted: in

Examining the cholesterol ion profiles showed that the positively charged molecular ion after the loss of its hydroxyl group $[M-OH]^+$ was seen to be most intense at the outer surface and decreased within the first 3 nm before reaching a relatively constant value in the interior of the particles (Figure 2c). This is consistent with cholesterol molecules sitting at the lipid surface with their hydroxyl groups oriented into the hydrophilic region occupied by the PEG, and would be consistent with there being areas with only a thin PEG coverage that allow this $[M-OH]^+$ ion to be present in the earliest analyses.

The intact sodiated molecular ion $[M+Na]^+$ appears to follow a profile which would be expected from an ion generated from the sum of the profiles containing sodium chloride (Na_2Cl^+) and cholesterol $[M-OH]^+$, at a low intensity at the outermost surface and increases to a plateau at approximately the same depth of 3 nm. Therefore, we interpret this as an analytical artefact rather than indicating the distribution of a molecular component in the LNP formulation. The cholesterol-Na adduct secondary ion may have formed at the surface during the collision cascade generating secondary ions from both of these species. This is in contrast to the PEG sodium adducts that we did not consider artefactual since their profile that could result from a combination of the profiles of the separate components (the sodium and the PEG).

Deleted: artifact

Deleted: -

Deleted: secondary ion

Deleted: considered

Deleted: it did not display a

Deleted: other

We note that previous work on pure cholesterol samples prepared without water and without the complicating presence of salts, showed depth profile data of the molecular ion after losing the hydroxyl group $[M-OH]^+$ at the outermost surface and the molecular ion $[M-H]^+$ below it, consistent with an inward molecular orientation of the hydroxyl group.²⁷ In this LNP system, we do not see an intense $[M-H]^+$ ion, most likely due to adduct formation from sodium present in the PBS. While these two systems are very different and cannot be compared directly, the observation of the different cholesterol fragments suggests molecular

Deleted: molecule

Deleted: ²⁵

Deleted: to

Deleted: of different

Deleted: suggesting

orientation in a model pure crystalline system. This is consistent with our observation of molecular orientation of the larger PMG-PEG2k using GCIB etching in this LNP system, suggesting that the orientation of small molecules such as cholesterol may be detectable in some systems.

Deleted: ¶

Figure 2 OrbiSIMS depth profiles of different component fragment ions in Formulation 1. a) DSPC, b) Dlin-MC3-DMA and c) Cholesterol. The intensity has been normalised to total ion count and the ions: $C_5H_6N_5^+$, $C_6H_{12}NO_2^+$, $C_{37}H_{67}^+$ and $C_{27}H_{45}^+$ are presented on secondary intensity scale (indicated by arrows) for clarity due to the difference in overall peak intensities.

Discussion

RNA-Lipid Nanoparticles

A state of the art in LNP structural determination method is small angle neutron scattering (SANS) with contrast-variation to give information on the 3D compositional distribution of different lipids and RNA.^{5,28,29} Small angle X-ray scattering (SAXS) is used to probe the internal phase of LNPs and the interactions between RNA and ionisable lipids in the LNP core.³⁰ NMR is also used to provide structural insights.¹⁷ The Cryo-OrbiSIMS depth profiling capability presented here is complementary to these techniques. It is a *de novo* approach that does not require a preconceived model, knowledge of the identity of the lipids or deuterium labelling of components. These attributes would be particularly useful in the analysis of organic nanoparticles comprising more complex component mixtures derived from natural products.³¹ This also makes it amenable to the analysis of real-world over model formulations requiring labelling. The analysis throughput can be of the order of 20 depth profiles per hour, making this compatible with high throughput LNP formulation structure-activity investigations and discovery campaigns.³²

Recent advancements in artificial intelligence (AI) that connect lipid chemistry to in vivo performance in high throughput LNP discovery campaigns can be enhanced by incorporating insights on surface structure and composition.³³ These advances in analytics and AI have the potential to facilitate LNP prediction based on formulation composition, which can subsequently inform in vivo efficacy and clinical effectiveness.

Gaining insight into the relative positions of each component within lipid nanoparticles (LNPs) is essential. Such knowledge can help clarify the intricate behaviour of LNPs and contribute to the design of more efficient and safer formulations. Furthermore, it can play a vital role in quality control during the scale-up of manufacturing processes, thereby enhancing the translation of LNPs from the laboratory to clinical applications. Examining the Dlin-MC3-DMA profile suggests that it is located at the surface, with molecular and fragment ions exhibiting a drop in signal intensity from the surface (Figure 2b). This is mirrored by fragments from all parts of the molecule, headgroup and fatty acid tails. Towards the core, these same ion intensities begin to increase, suggesting that it is also at an elevated amount in the core of the particle. Notably, the depth in the profile where the intensity decreases, is at

Formatted: Font: Bold

Deleted: $C_5H_{15}NO_4P^+$, $C_{44}H_{89}NO_8P^+$, $C_{43}H_{80}NO_2^+$, $C_6H_{14}NO_2^+$,...

Deleted: *, Na_2Cl

Deleted: ¶

.....Page Break.....

Deleted: ^{26,27}

Deleted: ²⁸

Deleted: can

Deleted: useful

Deleted:

Formatted: Not Superscript/ Subscript

Formatted: Font: Italic

Deleted: more complex

Deleted: .

Deleted: ²⁹

Deleted: rather than labelled

Deleted: .

Deleted: ³⁰

Deleted: ³¹

Deleted: Additionally, when paired with experimental biophysical models that examine the components involved in LNP escape from the endosome, this approach can provide a powerful framework for studying mechanisms that could lead to improved therapeutic formulations.

Deleted: ³²

Deleted: decrease

the same position that the fatty acid intensities increase in the DMG PEG molecule, suggesting the presence of a potential hydrophobic barrier layer that could be envisaged as separating molecules like the ionisable lipid from the particle core. Molecules will partition based on their molecular properties, such as charge state and hydrophilicities, as exhibited at the pH of particle formulation in the formation of this hydrophobic barrier layer. As the molecular assembly of LNPs contributes to the determination of LNP structure and thereby function, further experimentation systematically varying the pH of formulation could enable the assessment of this hydrophobic barrier and its impact on molecular distribution.

Development of Cryo-OrbiSIMS method

The biointerface at the surface of LNPs, comprising the surface lipids identified here and bioadsorbate molecules from the biological milieu in service, is thin and can therefore be regarded as an ideal opportunity for calibration of SIMS for quantification since the matrix effect may be controllable. **It has been has shown that** in model organic systems the matrix effect is insignificant for thickness < 5 nm.²⁵ Furthermore, undertaking the analysis in **frozen hydrated state** is analogous to implantation of oxygen in semiconductors where SIMS is routinely used to quantify dopants, by implanting oxygen to 'normalise' the matrix, providing opportunity for quantification in future.²⁶ Importantly, **the presence of frozen hydrated water**, also increases the sensitivity to many species compared to dehydrated samples by up to **×10,000**.³⁵ Using standard mixtures to map the relationship between Orbitrap counts, secondary ion yield (counts/nm³)^{36,37} and molar molecular concentrations, may allow biointerface molecular quantification, so long as sufficiently large sets of calibration data can be generated and appropriately analysed to produce useful sensitivity factors and to monitor for matrix effects. Control of salts and characterisation of their effects in such calibration datasets will be important since they are unavoidable for real-world samples in vitro and in vivo.

Depth profiles of organic delta layer standards show that the depth accuracy and resolution is dependent on the secondary ions used as well as the primary ions and their energy.³⁸ Without standard samples comprising molecules from the LNPs, it is difficult to differentiate broadening due to sputter effects from actual concentration gradients representing component intermixing or heterogeneous particle chemistries. Therefore, the depth resolution expected on these spherical, real-world frozen hydrated LNP samples is difficult to estimate, **although sputter profiling of nano particulate geometries has been studied for well-defined gold and silica particles where the degradation of the depth content was quantified, this is likely not to transfer directly to the organic self-assembled particles considered here.**³⁹ We have therefore been cautious in our **quantitative** interpretation of the LNP **depth** profiles.

Deleted: ³³

Deleted: ice

Deleted: ³⁴

Deleted: ice

Formatted: Superscript

Deleted: x10

Deleted: -

Deleted: .

While standards for depth profile resolution estimation for LNP components are not readily available, supported lipid bi-layers **could be envisaged** which would have the advantage of being flat, or **multi-component micelles**, which would well represent the particulate geometry and approximate dimensions. This would allow the variability in PEG overlayer thickness inferred from the broadness of profiles seen herein to be ascribed to actual or experimental depth resolution limits. **The new analytical insights this novel method can achieve can be paired with endosomal mimics to examine the influence on LNP components in endosomal escape, offering a powerful framework for to improve therapeutic formulations.**³⁴³⁴

Deleted: these could be envisaged to be

Deleted: single- and mixed

In summary, we find that Cryo-OrbiSIMS depth profiling offers a valuable methodology for label free characterisation of the native LNP surface molecular structure **and stratification** in the frozen hydrated state. **In addition to this high-fidelity insight into the surface molecular stratification of the individual components, the orientation of the PEG-lipid can be identified, consistent with models from neutron scattering and well-established models of similar LNPs.** It is anticipated that this approach will find **application** in the rapidly **developing** RNA-LNP therapeutics space, as well as have utility more broadly in the organic biomaterials space where capturing native hydrated surface chemistry and **characterising its molecular constituents and their orientation is critical for engineering their performance.**

Deleted: ,

Deleted: -

Deleted: development

Methods

LNP Preparation

LNPs were formulated via microfluidic mixing using a Precision Nanosystems NanoAssemblr Ignite instrument with a formulation were composed of Dlin-MC3-DMA/DSPC/DMG-PEG2k/Cholesterol at a molar ratio of (50/10/1.5/38.5) with an N:P ratio of 6. Lipids were solubilized in ethanol. The RNA cargo was comprised of a 1:1 molar ratio of mRNA reporter cargo and formulated in a 50mM citrate buffer at pH 3.0. After mixing, nanoparticles were transferred to Slide-A-lyzer G2 dialysis cassettes (10k MWCO) and dialyzed into PBS in at least 200x excess buffer for two hours at room temperature followed by overnight dialysis with gentle stirring in refreshed buffer at 4°C. LNP size and polydispersity were determined using a Zetasizer Ultra Malvern Panalytical instrument and are presented in Figure 1a. LNP encapsulation efficiencies were evaluated using the Invitrogen™ Quant-it™ RiboGreen Reagent and RNA Assay Kit (Thermo Fisher).

Deleted: preparation

Deleted:

Deleted: assay kit.

Cryo-TEM

Quantifoil R2/1 Cu 300 TEM grids (Agar Scientific Ltd, Essex UK) were glow discharged prior to use to render them hydrophilic. LNP suspensions were automatically blotted with contact between the TEM grid surface and filter paper was pre-set to 3 or 6 s. They were then frozen in vitrified ice on these grids in liquid ethane using a Leica EM GP2 automatic plunge freezer and maintained under liquid nitrogen until required for analysis. A blotting time of 6s yielded satisfactory results for samples the 4mg/mL concentration samples presented in this paper.

Formatted: English (UK)

Samples were maintained at cryogenic temperatures during transfer and analysis using an Elsa™ cryo-transfer holder and temperature controller (Gatan). Imaging was performed on a JEOL 2100F FEG-TEM operating at 200kV and recorded on a K3 direct electron detection camera (Gatan), nominal defocus was applied and an objective aperture inserted to improve image quality. A sample temperature of -172°C ($\pm 0.5^{\circ}\text{C}$) was maintained during imaging. Images in this document were binned and converted to JPEG format to improve clarity, raw data is available online.

Cryo-OrbiSIMS Analysis

The samples post Cryo-TEM were retained in frozen hydrated state and transferred into the Leica VCM bath filled with liquid nitrogen. The samples were fixed to the Leica cryogenic block and introduced into the airlock on the cryo-sample holder via the Leica vacuum

Deleted: analysis

Deleted: kept

Deleted: were

Deleted: cryostage

transfer system. The analysis was carried out at -170°C using a closed-loop liquid nitrogen pumping system (IONTOF GmbH).

For the acquisition of the OrbiSIMS depth profiles, a 20 keV Ar_{3000}^{+} analysis beam of 20 μm diameter, was used as primary ion beam. The distribution is shown in Figure S5. OrbiSIMS profiles were acquired from 3 replicates from the same LNP batch. Ar_{3000}^{+} with duty cycle set to 4.2% and GCIB current was 220 pA. The Q-Exactive depth profile was run on the area of 400 ~~μm~~ \times 400 μm using random raster mode with crater size 498 \times 498 μm . The cycle time was set to 400 μs . Optimal target potential was set to -57 V in negative polarity and +57 V in positive polarity. Argon gas flooding was in operation in order to aid charge compensation, pressure in the main chamber was maintained at 9.0×10^{-7} ~~mbar~~. The spectra were collected in positive polarity, in mass range m/z 75–1125. The injection time was set to 500 ms and each area analysed lasted 1000 scans, the total ion dose per measurement was 2.62×10^{14} ions/ cm^2 . The static limit of primary ion dose, commonly used for primary beams which result in the build-up of sample damage (10^{12} ions/ cm^2) is reached at the third scan. Mass-resolving power was set to 240,000 at m/z 200. Etching removes the particle strata which are sampled sequentially as primary ion bombardment proceeds, with the exception of shadowing and preferential sputter yields discussed in the Yang et al. ³⁹

Deleted: ×

Deleted: bar

Calibration of the Orbitrap analyser was performed on the silver sample, using silver clusters following the method described by Passarelli *et al.* using the Bi_1^{+} liquid metal ion gun as a primary ion beam. ⁴⁰

We have used information from a published depth profiling organic molecular standard (Irganox) to estimate the depth from the primary ion dose, which is displayed as a secondary x axis (red) in Figure 1b. ⁴¹ The sputter yield from 20keV Ar_{3000}^{+} on Irganox was corrected for the reduced temperature (-170°C) used in this work compared to the literature values ⁴² using the calculations developed by Seah *et al.*

Deleted: ³⁹

Deleted: ⁴⁰

Each depth profile was acquired from the full thickness of the sample. An initial nanoparticle components region was first seen, followed by a complex mixture of inorganic ions (not shown) interpreted to be from the surface of the carbon support substrate, were found, appearing in the profile below the particles before the substrate was reached. We do not interpret the relative positions of these ions, instead focussing on the organics which represent the lipids.

Deleted: and a

Deleted: , salts

Deleted: , although we do discuss the role of inorganic ions in secondary ion enhancement and suppression

While depth profiling is commonly used in SIMS for the analysis of layered systems,⁴³ it is limited to assessing depth distributions below half of the particle diameter due to the cumulative development of topography in the etching process.⁴⁴ Therefore the data in this paper that is interpreted is limited to a primary ion dose of 3×10^{13} ions/cm².

Deleted: ⁴¹

Deleted: ⁴²

Time of flight secondary ion mass spectrometry (ToF SIMS)

Deleted:

Formatted: Font: Not Bold

For the acquisition of liquid metal ion gun (LMIG) ToF-SIMS spectra of the lipid nanoparticle formulation, a 30 keV Bi₃⁺ primary beam was used. LMIG current was 0.03 pA. The acquisition was run on the area of 500 μm × 500 μm using random raster mode. Optimal target potential was set to +58 V. The total ion dose per measurement was 7.36×10^9 . The analysis was carried out at -170°C using a closed-loop liquid nitrogen pumping system (IONTOF GmbH).

Acknowledgements

The OrbiSIMS facility was funded by the EPSRC Strategic Equipment Grant '3D OrbiSIMS: Label free chemical imaging of materials, cells and tissues' EP/P029868/1. The ~~CryoTEM~~ holder and detector were funded by the EPSRC Research Grant 'A New Correlative Approach for Structure Determination & Imaging of Molecular Materials' EP/W006413/1. Reece Franklin of University of Nottingham assisted with cryogenic sample preparation of replicate analyses. Kerry Benenato of Sail Biomedicines read and commented on the manuscript during its drafting.

Data Availability

Data include in this manuscript are available at <https://rdmc.nottingham.ac.uk/>

Deleted: cryoTEM

Table

Deleted: Tables

Cholesterol	 The chemical structure of cholesterol is shown, featuring a four-ring steroid nucleus, a hydroxyl group at C3, two methyl groups at C10 and C13, and a branched hydrocarbon side chain at C17.
Dlin-MC3-DMA	 The chemical structure of Dlin-MC3-DMA is shown, consisting of a long hydrocarbon chain with three double bonds and a dimethylammonium group attached to the chain.
DSPC	 The chemical structure of DSPC is shown, featuring two long hydrocarbon tails, a phosphate group, and a dimethylammonium group.
DMG-PEG2K	 The chemical structure of DMG-PEG2K is shown, consisting of two long hydrocarbon tails, a phosphate group, and a polyethylene glycol (PEG) chain with 44 repeating units.

Table 1 Individual component structures.

Supplementary Figures

Deleted: <object>a)¶

Formatted Table

Figure S1 Characterization of LNPs. a) z-average size (nm), PDI, and encapsulation efficiency (%EE) as described in the experimental methods. b) DLS size distribution curves. **c) Frozen nanoparticle schematic for the Cryo-OrbiSIMS analysis.** **d) LMIG ToF spectra of the LNP formulation acquired in cryogenic conditions. The repetitive pattern represents water cluster peaks, the first 3 water cluster peaks are assigned in the spectrum, with the other indicated by asterisk.**

Figure S2 Positive polarity *Cryo*-OrbiSIMS spectra of individual lipid component samples (m/z 75–1125). All spectra are normalised to total ion count and presented on individual intensity scales for clarity.

Figure S3 Cryo-OrbiSIMS profiles of Formulation 1 displaying profiles from RNA fragments from adenine, guanine and cytosine along with the PEG ions. Uracil was not seen in the spectra. The intensities have been normalised to total ion count and all RNA fragment ions are presented on secondary intensity scale for clarity.

Figure S4 Cryo-OrbiSIMS profiles acquired from on different samples from the same batch of LNPs on different occasions as the data in Figure 1b. The intensity has been normalised to total ion count and the summed **FA14:Q** ions are presented on secondary intensity scale for both depth profiles for clarity.

Deleted: PEG

Figure S5 Cluster size distribution of the Ar_{3000}^+ primary ion beam used for Cryo-OrbiSIMS analysis.

Table S1, DMG-PEG2k fragments containing both DMG and PEG fragments. The PEG component, which were summed in the depth profiles.

m/z	SIMS assignment	Atomic composition	Structural formula
227.2016	C₁₄H₂₇O₂⁺	H(CH₂)₁₃C(=O)O⁺	[FA(14:0)]⁺
251.1983	C₁₄H₂₈O₂Na⁺	H(CH₂)₁₃C(=O)ONa⁺	[FA(14:0)]Na⁺
493.2618	C ₂₁ H ₄₂ O ₁₁ Na ⁺	C_{2x+1}H_{4x+2}O_{x+1}Na⁺	Na[O-CH₂-CH₂]_x-O-CH₂⁺
495.4408	C ₃₁ H ₅₉ O ₄ ⁺	DMG⁺	[FA(14:0)]₂C₃H₅⁺
507.2775	C ₂₂ H ₄₄ O ₁₁ Na ⁺	C_{2x}H_{4x}O_xNa⁺	Na[O-CH₂-CH₂]_x⁺
537.2881	C ₂₃ H ₄₆ O ₁₂ Na ⁺	C_{2x+1}H_{4x+2}O_{x+1}Na⁺	Na[O-CH₂-CH₂]_x-O-CH₂⁺
551.3037	C ₂₄ H ₄₈ O ₁₂ Na ⁺	C_{2x}H_{4x}O_xNa⁺	Na[O-CH₂-CH₂]_x⁺
565.3193	C ₂₅ H ₅₀ O ₁₂ Na ⁺	C_{2x+3}H_{4x+6}O_{x+1}Na⁺	NaCH₂[O-CH₂-CH₂]_x-O-CH₂-CH₂⁺
567.2985	C ₂₄ H ₄₈ O ₁₃ Na ⁺	C_{2x}H_{4x}O_{x+1}Na⁺	Na[O-CH₂-CH₂]_xO⁺
581.3142	C ₂₅ H ₅₀ O ₁₃ Na ⁺	C_{2x+1}H_{4x+2}O_{x+1}Na⁺	Na[O-CH₂-CH₂]_x-O-CH₂⁺
609.3454	C ₂₇ H ₅₄ O ₁₃ Na ⁺	C_{2x+3}H_{4x+6}O_{x+1}Na⁺	NaCH₂[O-CH₂-CH₂]_x-O-CH₂-CH₂⁺
625.3403	C ₂₇ H ₅₄ O ₁₄ Na ⁺	C_{2x+1}H_{4x+2}O_{x+1}Na⁺	Na[O-CH₂-CH₂]_x-O-CH₂⁺
653.3717	C ₂₉ H ₅₈ O ₁₄ Na ⁺	C_{2x+3}H_{4x+6}O_{x+1}Na⁺	NaCH₂[O-CH₂-CH₂]_x-O-CH₂-CH₂⁺
669.3665	C ₂₉ H ₅₈ O ₁₅ Na ⁺	C_{2x+1}H_{4x+2}O_{x+1}Na⁺	Na[O-CH₂-CH₂]_x-O-CH₂⁺
697.3978	C ₃₁ H ₆₂ O ₁₅ Na ⁺	C_{2x+3}H_{4x+6}O_{x+1}Na⁺	NaCH₂[O-CH₂-CH₂]_x-O-CH₂-CH₂⁺
713.3928	C ₃₁ H ₆₂ O ₁₆ Na ⁺	C_{2x+1}H_{4x+2}O_{x+1}Na⁺	Na[O-CH₂-CH₂]_x-O-CH₂⁺
741.424	C ₃₃ H ₆₆ O ₁₆ Na ⁺	C_{2x+3}H_{4x+6}O_{x+1}Na⁺	NaCH₂[O-CH₂-CH₂]_x-O-CH₂-CH₂⁺
757.4189	C ₃₃ H ₆₆ O ₁₇ Na ⁺	C_{2x+1}H_{4x+2}O_{x+1}Na⁺	Na[O-CH₂-CH₂]_x-O-CH₂⁺
785.4502	C ₃₅ H ₇₀ O ₁₇ Na ⁺	C_{2x+3}H_{4x+6}O_{x+1}Na⁺	NaCH₂[O-CH₂-CH₂]_x-O-CH₂-CH₂⁺
801.4452	C ₃₅ H ₇₀ O ₁₈ Na ⁺	C_{2x+1}H_{4x+2}O_{x+1}Na⁺	Na[O-CH₂-CH₂]_x-O-CH₂⁺
829.4766	C ₃₇ H ₇₄ O ₁₈ Na ⁺	C_{2x+3}H_{4x+6}O_{x+1}Na⁺	NaCH₂[O-CH₂-CH₂]_x-O-CH₂-CH₂⁺
845.4713	C ₃₇ H ₇₄ O ₁₉ Na ⁺	C_{2x+1}H_{4x+2}O_{x+1}Na⁺	Na[O-CH₂-CH₂]_x-O-CH₂⁺
873.5026	C ₃₉ H ₇₈ O ₁₉ Na ⁺	C_{2x+3}H_{4x+6}O_{x+1}Na⁺	NaCH₂[O-CH₂-CH₂]_x-O-CH₂-CH₂⁺
889.4975	C ₃₉ H ₇₈ O ₂₀ Na ⁺	C_{2x+1}H_{4x+2}O_{x+1}Na⁺	Na[O-CH₂-CH₂]_x-O-CH₂⁺
917.5291	C ₄₁ H ₈₂ O ₂₀ Na ⁺	C_{2x+3}H_{4x+6}O_{x+1}Na⁺	NaCH₂[O-CH₂-CH₂]_x-O-CH₂-CH₂⁺
933.5238	C ₄₁ H ₈₂ O ₂₁ Na ⁺	C_{2x+1}H_{4x+2}O_{x+1}Na⁺	Na[O-CH₂-CH₂]_x-O-CH₂⁺
961.5549	C ₄₃ H ₈₆ O ₂₁ Na ⁺	C_{2x+3}H_{4x+6}O_{x+1}Na⁺	NaCH₂[O-CH₂-CH₂]_x-O-CH₂-CH₂⁺
977.5498	C ₄₃ H ₈₆ O ₂₂ Na ⁺	C_{2x+1}H_{4x+2}O_{x+1}Na⁺	Na[O-CH₂-CH₂]_x-O-CH₂⁺
1005.581	C ₄₅ H ₉₀ O ₂₂ Na ⁺	C_{2x+3}H_{4x+6}O_{x+1}Na⁺	NaCH₂[O-CH₂-CH₂]_x-O-CH₂-CH₂⁺
1021.576	C ₄₅ H ₉₀ O ₂₃ Na ⁺	C_{2x+1}H_{4x+2}O_{x+1}Na⁺	Na[O-CH₂-CH₂]_x-O-CH₂⁺
1049.608	C ₄₇ H ₉₄ O ₂₃ Na ⁺	C_{2x+3}H_{4x+6}O_{x+1}Na⁺	NaCH₂[O-CH₂-CH₂]_x-O-CH₂-CH₂⁺
1065.602	C ₄₇ H ₉₄ O ₂₄ Na ⁺	C_{2x+1}H_{4x+2}O_{x+1}Na⁺	Na[O-CH₂-CH₂]_x-O-CH₂⁺

Deleted: Supplementary

Deleted: 1

Deleted: only the

Formatted Table

Formatted Table

1093.633	$C_{49}H_{98}O_{24}Na^+$	$C_{2x+3}H_{4x+6}O_{x+1}Na^+$	$NaCH_2[O-CH_2-CH_2]_x-O-CH_2-CH_2^+$
1109.628	$C_{49}H_{98}O_{25}Na^+$	$C_{2x+1}H_{4x+2}O_{x+1}Na^+$	$Na[O-CH_2-CH_2]_x-O-CH_2^+$

Formatted: Polish

References

1. Hou, X., Zaks, T., Langer, R. & Dong, Y. Lipid nanoparticles for mRNA delivery. *Nat Rev Mater* **6**, 1078–1094 (2021).
2. Dilliard, S. A., Cheng, Q. & Siegwart, D. J. On the mechanism of tissue-specific mRNA delivery by selective organ targeting nanoparticles. *Proc Natl Acad Sci U S A* **118**, (2021).
3. Francia, V. *et al.* A magnetic separation method for isolating and characterizing the biomolecular corona of lipid nanoparticles. *Proceedings of the National Academy of Sciences* **121**, (2024).
4. Tam, Y. Y. C., Chen, S. & Cullis, P. R. Advances in Lipid Nanoparticles for siRNA Delivery. *Pharmaceutics* **5**, 498–507 (2013).
5. Arteta, M. Y. *et al.* Successful reprogramming of cellular protein production through mRNA delivered by functionalized lipid nanoparticles. *Proc Natl Acad Sci U S A* **115**, E3351–E3360 (2018).
6. Blakney, A. K., McKay, P. F., Yus, B. I., Aldon, Y. & Shattock, R. J. Inside out: optimization of lipid nanoparticle formulations for exterior complexation and in vivo delivery of saRNA. *Gene Therapy* *2019 26:9* **26**, 363–372 (2019).
7. Kulkarni, J. A. *et al.* Spontaneous, solvent-free entrapment of siRNA within lipid nanoparticles †. *Nanoscale* **12**, 23959 (2020).
8. Tenchov, R., Bird, R., Curtze, A. E. & Zhou, Q. Lipid Nanoparticles from Liposomes to mRNA Vaccine Delivery, a Landscape of Research Diversity and Advancement. *ACS Nano* **15**, 16982–17015 (2021).
9. Hamilton, A. G., Swingle, K. L. & Mitchell, M. J. Biotechnology: Overcoming biological barriers to nucleic acid delivery using lipid nanoparticles. *PLoS Biol* **21**, e3002105 (2023).
10. Wang, M. M. *et al.* Elucidation of lipid nanoparticle surface structure in mRNA vaccines. *Scientific Reports* *2023 13:1* **13**, 1–8 (2023).
11. Gref, R. *et al.* Biodegradable Long-Circulating Polymeric Nanospheres. *Science (1979)* **263**, 1600–1603 (1994).
12. Tenchov, R., Bird, R., Curtze, A. E. & Zhou, Q. Lipid Nanoparticles from Liposomes to mRNA Vaccine Delivery, a Landscape of Research Diversity and Advancement. *ACS Nano* **15**, 16982–17015 (2021).
13. Heyes, J., Hall, K., Tailor, V., Lenz, R. & MacLachlan, I. Synthesis and characterization of novel poly(ethylene glycol)-lipid conjugates suitable for use in drug delivery. *Journal of Controlled Release* **112**, 280–290 (2006).
14. Kulkarni, J. A. *et al.* On the Formation and Morphology of Lipid Nanoparticles Containing Ionizable Cationic Lipids and siRNA. *ACS Nano* **12**, 4787–4795 (2018).
15. Eygeris, Y., Patel, S., Jozic, A., Sahay, G. & Sahay, G. Deconvoluting Lipid Nanoparticle Structure for Messenger RNA Delivery. *Nano Lett* **20**, 4543–4549 (2020).
16. Philipp, J. *et al.* pH-dependent structural transitions in cationic ionizable lipid mesophases are critical for lipid nanoparticle

Formatted: Indent: Left: 3.39 cm

- function. *Proceedings of the National Academy of Sciences* **120**, (2023).
17. Wang, M. M. *et al.* Elucidation of lipid nanoparticle surface structure in mRNA vaccines. *Sci Rep* **13**, 16744 (2023).
 18. Viger-Gravel, J. *et al.* Structure of Lipid Nanoparticles Containing siRNA or mRNA by Dynamic Nuclear Polarization-Enhanced NMR Spectroscopy. *Journal of Physical Chemistry B* **122**, 2073–2081 (2018).
 19. Cant, D. J. H. *et al.* Cryo-XPS for Surface Characterization of Nanomedicines. *J Phys Chem A* **127**, 8220–8227 (2023).
 20. Bamford, S. E. *et al.* High resolution imaging and analysis of extracellular vesicles using mass spectral imaging and machine learning. *Journal of Extracellular Biology* **2**, (2023).
 21. Lovrić, J. *et al.* Analysis of liposome model systems by time-of-flight secondary ion mass spectrometry. *Surface and Interface Analysis* **46**, 74–78 (2014).
 22. Passarelli, M. K. *et al.* The 3D OrbiSIMS - Label-free metabolic imaging with subcellular lateral resolution and high mass-resolving power. *Nat Methods* **14**, 1175–1183 (2017).
 23. Yu, M., Kang, X. & Qian, L. Interactions between monovalent cations and polyethylene glycol: A study at micro level. *Colloids Surf A Physicochem Eng Asp* **680**, 132731 (2024).
 24. Fujii, M., Shishido, R., Satoh, T., Suzuki, S. & Matsuo, J. Effects of molecular weight and cationization agent on the sensitivity of Bi cluster secondary ion mass spectrometry. *Rapid Communications in Mass Spectrometry* **30**, 1722–1726 (2016).
 25. ~~Shard, A. G., Spencer, S. J., Smith, S. A., Havelund, R. & Gilmore, I. S. The matrix effect in organic secondary ion mass spectrometry. *Int J Mass Spectrom* **377**, 599–609 (2015).~~
 26. ~~Seah, M. P., Havelund, R., Spencer, S. J. & Gilmore, I. S. Quantifying SIMS of Organic Mixtures and Depth Profiles—Characterizing Matrix Effects of Fragment Ions. *J Am Soc Mass Spectrom* **30**, 309–320 (2019).~~
 27. Rakowska, P. D., Seah, M. P., Vorng, J. L., Havelund, R. & Gilmore, I. S. Determination of the sputtering yield of cholesterol using Ar: N⁺ and C60⁽⁺⁾ cluster ions. *Analyst* **141**, 4893–4901 (2016).
 28. ~~Gilbert, J. *et al.* Evolution of the structure of lipid nanoparticles for nucleic acid delivery: From in situ studies of formulation to colloidal stability. *J Colloid Interface Sci* **660**, 66–76 (2024).~~
 29. ~~Sebastiani, F. *et al.* Apolipoprotein E Binding Drives Structural and Compositional Rearrangement of mRNA-Containing Lipid Nanoparticles. *ACS Nano* **15**, 6709–6722 (2021).~~
 30. ~~Philipp, J. *et al.* pH-dependent structural transitions in cationic ionizable lipid mesophases are critical for lipid nanoparticle function. *Proceedings of the National Academy of Sciences* **120**, (2023).~~
 31. ~~Iravani, S. & Varma, R. S. Plant-Derived Edible Nanoparticles and miRNAs: Emerging Frontier for Therapeutics and Targeted Drug-Delivery. *ACS Sustain Chem Eng* **7**, 8055–8069 (2019).~~

Moved (insertion) [2]

Deleted: 25.

Moved (insertion) [3]

Deleted: 26

Deleted: 27

Deleted: 28

Deleted: 29

32. Kanasty, R., Dorkin, J. R., Vegas, A. & Anderson, D. Delivery materials for siRNA therapeutics. *Nature Materials* 2013 12:11 12, 967–977 (2013).
33. Witten, J. *et al.* Artificial intelligence-guided design of lipid nanoparticles for pulmonary gene therapy. *Nature Biotechnology* (in press) (2024).
34. Aliakbarinodehi, N. *et al.* Time Resolved Inspection of Ionizable-Lipid Facilitated Lipid Nanoparticle Disintegration and Cargo Release at an Endosomal Membrane Mimic. *ACS Nano* 22989–23000 (2024) doi:10.1101/2024.02.22.580934.
35. Zhang, J. *et al.* Cryo-OrbiSIMS for 3D Molecular Imaging of a Bacterial Biofilm in Its Native State. *Anal Chem* 92, 9008–9015 (2020).
36. Gilmore, I. & Keenan, M. Orbitrap noise structure and method for noise-unbiased multivariate analysis. *Preprint (responding to reviewers)* (2024) doi:10.21203/RS.3.RS-3911895/V1.
37. Zhou, Y. *et al.* OrbiSIMS depth profiling of semiconductor materials—Useful yield and depth resolution. *Journal of Vacuum Science & Technology A* 42, (2024).
38. Havelund, R., Seah, M. P., Tiddia, M. & Gilmore, I. S. SIMS of Organic Materials—Interface Location in Argon Gas Cluster Depth Profiles Using Negative Secondary Ions. *J Am Soc Mass Spectrom* 29, 774–785 (2018).
39. Yang, L. *et al.* Depth Profiling and Melting of Nanoparticles in Secondary Ion Mass Spectrometry (SIMS). *The Journal of Physical Chemistry C* 117, 16042–16052 (2013).
40. Passarelli, M. K. *et al.* The 3D OrbiSIMS - Label-free metabolic imaging with subcellular lateral resolution and high mass-resolving power. *Nat Methods* 14, 1175–1183 (2017).
41. Seah, M. P. Universal Equation for Argon Gas Cluster Sputtering Yields. *The Journal of Physical Chemistry C* 117, 12622–12632 (2013).
42. Seah, M. P., Havelund, R. & Gilmore, I. S. Systematic Temperature Effects in the Argon Cluster Ion Sputter Depth Profiling of Organic Materials Using Secondary Ion Mass Spectrometry. *J Am Soc Mass Spectrom* 27, 1411–1418 (2016).
43. Bailey, J. *et al.* 3D ToF-SIMS Imaging of Polymer Multilayer Films Using Argon Cluster Sputter Depth Profiling. 3–8 (2015) doi:10.1021/am507663v.
44. Seah, M. P. Topography effects and monatomic ion sputtering of undulating surfaces, particles and large nanoparticles: Sputtering yields, effective sputter rates and topography evolution. *Surface and Interface Analysis* 44, 208–218 (2012).

Deleted: 30

Deleted: 31

Deleted: 32

Moved up [2]: → Shard, A. G., Spencer, S. J., Smith, S. A., Havelund, R. & Gilmore, I. S. The matrix effect in organic secondary ion mass spectrometry. *Int J Mass Spectrom* 377, 599–609 (2015).[†]

Moved up [3]: → Seah, M. P., Havelund, R., Spencer, S. J. & Gilmore, I. S. Quantifying SIMS of Organic Mixtures and Depth Profiles—Characterizing Matrix Effects of Fragment Ions. *J Am Soc Mass Spectrom* 30, 309–320 (2019).[†]

Deleted: 33.

Deleted: 34.

Formatted: Indent: Left: 3.38 cm

Deleted: 39.

Formatted: Indent: Left: 3.39 cm

Deleted: 40

Deleted: 41

Deleted: 42